# ELIMINATING STEADY-STATE OSCILLATIONS IN DISTRIBUTED OPTIMIZATION AND LEARNING VIA ADAPTIVE STEPSIZE

## ABSTRACT

Distributed stochastic optimization and learning is gaining increasing traction due to its ability to enable large-scale data processing and model training across multiple agents without the need for centralized coordination. However, existing distributed stochastic optimization and learning approaches, such as distributed SGD and their variants, generally face a dilemma in stepsize selection: a small stepsize leads to low convergence speed, whereas a large stepsize often incurs pronounced steady-state oscillations, which prevents the algorithm from achieving stable convergence accuracy. In this paper, we propose an adaptive stepsize approach for distributed stochastic optimization and learning that can eliminate steady-state oscillations and ensure fast convergence. Such guarantees are unattained by existing adaptive stepsize approaches, even in centralized optimization and learning. We prove that our proposed algorithm achieves linear convergence with respect to the iteration number, and that the convergence error decays sublinearly with the batch size of sampled data points. In the specific case in terms of deterministic distributed optimization with exact gradients accessible to agents, we prove that our proposed algorithm linearly converges to an exact optimal solution. Moreover, we quantify that the computational complexity of the proposed algorithm is on the order of $\mathcal{O}(\log(\epsilon^{-1}))$, which matches the existing results on adaptive stepsize approaches for centralized optimization/learning. Experimental results on machine learning benchmarks confirm the effectiveness of our proposed approach.

## 1 INTRODUCTION

With the advance of the era of big data, distributed stochastic optimization and learning methods have attracted increasing attention due to their unique ability to leverage the computational power of multiple devices to accelerate training (Nedic & Ozdaglar, 2009; Yang & Johansson, 2010; Shamir & Srebro, 2014; Lian et al., 2017; Nedić & Liu, 2018; Yang et al., 2019; Kim et al., 2024; Hu et al., 2024). Unlike centralized optimization and learning methods (Wang & Elia, 2011; Andrychowicz et al., 2016; Ruder, 2016) that typically rely on a central server to aggregate local model parameters or data from all participating agents, distributed methods allow each agent to collaboratively learn a global model using only its own local dataset and information exchanged with neighboring agents, without the assistance of any centralized server or aggregator (Scaman et al., 2018; Liu et al., 2020; Yang et al., 2022).

However, existing distributed stochastic optimization/learning approaches often face a dilemma in stepsize selection (Jacobs, 1988; Schaul et al., 2013; Wei et al., 2020; Zhuang et al., 2020; Li et al., 2024a; Huang et al., 2024b; Crawshaw et al., 2025). Specifically, an excessively small stepsize may lead to an overly low convergence speed (Srivastava & Nedic, 2011; Lin et al., 2023; Sharifnassab et al., 2024), whereas an excessively large stepsize often causes pronounced steady-state oscillations or overshoot, which prevents the algorithm from achieving stable convergence accuracy (Andriushchenko et al., 2023; Huang et al., 2024a). Recently, several adaptive or automatic stepsize approaches have been proposed for centralized optimization and learning (Fletcher, 2005; Kingma, 2014; Rolinek & Martius, 2018; Li & Orabona, 2019; Malitsky & Mishchenko, 2019;

Kavis et al., 2022; Jiang & Stich, 2023; Malitsky & Mishchenko, 2024). However, these approaches generally rely on a centralized server to coordinate computation that are impractical in a fully distributed setting where no centralized server/aggregator exists to determine a common stepsize across all agents (Nedić et al., 2018). Although some works have attempted to extend adaptive stepsize approaches to distributed optimization and learning (Nazari et al., 2022; Carnevale et al., 2022; Xie et al., 2022; Ramezani-Kebrya et al., 2023; Chen & Wang, 2024; Kuruzov et al., 2024; Saravanos et al., 2024), most of them still either require a centralized server to collect local model parameters/stepsizes from all agents (Ramezani-Kebrya et al., 2023; Chen & Wang, 2024; Kuruzov et al., 2024), or are limited to scenarios where agents must have access to accurate gradients of the objective functions (Carnevale et al., 2022; Xie et al., 2022; Saravanos et al., 2024) for stepsize adjustment. The only exception is the work in Nazari et al. (2022), which achieves adaptive stepsize adjustments in distributed online learning by normalizing the gradient using an accumulated sum of historical gradient values. However, this approach leads to a rapidly decaying stepsize, which in turn results in slow convergence in the later stages of the algorithm (see our experimental results in Fig. 5 in Appendix C.3 for details). To the best of our knowledge, no existing adaptive stepsize approaches can ensure fast and stable convergence in fully distributed stochastic optimization/learning.

Our contributions are summarized as follows:

1. We propose an adaptive stepsize algorithm for fully distributed stochastic optimization and learning. This is in stark contrast to existing adaptive stepsize approaches, which either rely on a centralized server to coordinate a common stepsize across all agents (in, e.g., Ramezani-Kebrya et al. (2023); Kim et al. (2024); Chen & Wang (2024); Kuruzov et al. (2024)), or require that agents have access to accurate gradients of the objective functions (Carnevale et al., 2022; Xie et al., 2022; Saravanos et al., 2024)—which, however, are often hard to obtain in real-world applications where the randomness in sampled data results in only noisy gradients being accessible to agents. To the best of our knowledge, this is the first adaptive (non-monotone decreasing) stepsize approach for fully distributed stochastic optimization/learning, without the need for accurate gradients or the assistance of any centralized servers.

2. Our adaptive stepsize algorithm can eliminate steady-state oscillations and ensure stable convergence accuracy in the later stages of the algorithm. This is unattained by most existing adaptive stepsize approaches even in centralized optimization and learning (Fletcher, 2005; Li & Orabona, 2019; Kavis et al., 2022; Jiang & Stich, 2023). The key enabler is our novel design of the stepsize update rule, which allows each agent to dynamically adjust its individual stepsizes based on locally estimated curvature of the global objective function. This provides each agent with large stepsizes in the early stages to accelerate convergence, and extremely small stepsizes near the global optimum to ensure stable convergence accuracy (see our experimental results in Figs. 1(d)-1(f) and Figs. 2(d)-2(f) for details). Furthermore, since stable convergence accuracy is achieved in the later stages of our algorithm, we can also provide a clear stopping criterion[1] for each agent in distributed optimization and learning, which is rarely addressed in the state-of-the-art literature.

3. In addition to eliminating steady-state oscillations, we also establish the convergence rate and computational complexity of our algorithm for both stochastic and deterministic distributed optimization and learning, which is different from existing adaptive stepsize results in, e.g., McMahan & Streeter (2014); Yang et al. (2019); Crawshaw et al. (2025) that focus solely on deterministic cases where accurate gradients of objective functions are accessible to agents. For distributed stochastic optimization/learning, we prove that our algorithm achieves linear convergence with respect to the number of algorithm iterations, and that the convergence error decays sublinearly with the batch size of sampled data points. For the deterministic case, we prove that our algorithm linearly converges to an exact optimal solution.

4. We systematically quantify that the computational complexity of our algorithm is on the order of $\mathcal{O}(\log(\epsilon^{-1}))$ for both stochastic and deterministic cases, which matches the existing results on adaptive stepsize approaches for centralized optimization and learning in, e.g., (Kavis et al., 2022; Yang & Ma, 2023).

5. We conduct experimental evaluations using several machine learning benchmark datasets, including the "MNIST" dataset, the "CIFAR-10" dataset, and the "CIFAR-100" dataset. The

---

[1]We use the "stopping criterion" to denote the condition that determines when each agent in a distributed stochastic optimization and learning algorithm terminates its iterations (Vlachos, 2008; Ding et al., 2025).

results confirm the effectiveness of our algorithm in terms of both test accuracy and steady-state convergence performance.

## 2 RELATED WORK

**Distributed stochastic optimization and learning.** Distributed stochastic optimization methods have been widely employed in modern machine learning (Yang, 2013; Xin et al., 2020; Nedic, 2020; Guo et al., 2020; Pu et al., 2020; Allen-Zhu et al., 2020; Khaled & Jin, 2023; Song et al., 2025). However, most existing methods require all agents to share a common stepsize that is either fixed (Pu & Nedić, 2021; Koloskova et al., 2021; Nguyen et al., 2023; Song et al., 2024) or diminishing (Jakovetic et al., 2018; Dieuleveut & Patel, 2019; Li et al., 2024b; Lee et al., 2025). The fixed stepsize causes pronounced overshoot or oscillations near the global optimal solution (Pu & Nedić, 2021; Koloskova et al., 2021; Nguyen et al., 2023), whereas diminishing stepsizes often lead to an overly low convergence speed, both of which prevent the algorithm from achieving stable convergence accuracy (as shown in our experimental results in Fig. 1 and Fig. 2). Given these limitations, designing an adaptive stepsize approach that allows each participating agent to adaptively adjust its individual stepsizes is a promising direction for improving convergence speed and ensuring stable learning performance in distributed stochastic optimization and learning.

**Adaptive stepsize approaches.** Several adaptive stepsize approaches have been proposed for centralized optimization and learning (Fletcher, 2005; Kingma, 2014; Rolinek & Martius, 2018; Li & Orabona, 2019; Malitsky & Mishchenko, 2019; Kavis et al., 2022; Jiang & Stich, 2023; Malitsky & Mishchenko, 2024). However, these methods typically consider a single agent setting where learning is performed with only one adaptive stepsize adjustment. This makes them inapplicable to fully distributed stochastic optimization and learning, where multiple agents cooperatively perform learning and each agent has its own stepsize updates. Moreover, the existing adaptive stepsize approaches often lead to steady-state oscillations, which prevent stable convergence accuracy in the later stages of the algorithm and hinder the determination of a clear stopping criterion (as shown in our experimental results in Fig. 2). Although some efforts have attempted to extend adaptive stepsize approaches to distributed optimization and learning (Nazari et al., 2022; Carnevale et al., 2022; Xie et al., 2022; Ramezani-Kebrya et al., 2023; Chen & Wang, 2024; Kuruzov et al., 2024; Saravanos et al., 2024), most of them still rely on a centralized server to collect local model parameters/stepsizes from all agents to coordinate a stepsize (Ramezani-Kebrya et al., 2023; Chen & Wang, 2024; Kuruzov et al., 2024), or are limited to scenarios where accurate gradients of the objective functions must be accessible to agents (Carnevale et al., 2022; Xie et al., 2022; Saravanos et al., 2024), both of which are impractical in a fully distributed and stochastic setting. The only exception is the recent work in Nazari et al. (2022), which achieves stepsize adjustments in distributed stochastic optimization and learning. However, its approach parallels adaptive gradient methods (e.g., ADAM in Kingma (2014)), which makes the stepsizes decay rapidly in practical neural-network training, thereby leading to a low convergence speed in the later stages of the algorithm (as shown in our experimental results in Fig. 5 in Appendix C.3). To the best of our knowledge, no adaptive stepsize approaches have been reported for distributed stochastic optimization and learning that can ensure both fast convergence and stable steady-state performance.

*Notations:* We use $\mathbb{R}^n$ to denote the $n$-dimensional real Euclidean space and $\mathbb{N}(\mathbb{N}^+)$ to denote the set of nonnegative (positive) integers. We write $\mathbf{0}_n$ and $\mathbf{1}_n$ for $n$-dimensional column vectors of all zeros and all ones, respectively; in both cases we suppress the dimension when clear from the context. We use $\langle x, y \rangle$ to denote the inner product of two vectors and $\| \cdot \|$ to denote the Euclidean norm of a vector. We write $\mathbb{E}[x]$ for the expected value of a random variable $x$. We use $[a]_+ = \max\{0, a\}$ to refer to the maximum of 0 and $a$ for any real number $a$ and the convention $\frac{a}{0} = +\infty$ for any $a > 0$. We denote the set of $m$ agents as $[m]$ and add an overbar to a letter to represent the average of $m$ agents, e.g., $\bar{x} = \frac{1}{m} \sum_{i=1}^{m} x_i$.

## 3 PROBLEM FORMULATION

We consider $m$ agents that cooperatively learn a common optimal model parameter $x^*$ to the following stochastic optimization problem (Sundhar Ram et al., 2010; Lian et al., 2017; Chen & Wang,

2024):

$$\min_{x \in \mathbb{R}^n} f(x) = \frac{1}{m} \sum_{i=1}^{m} f_i(x), \quad f_i(x) = \mathbb{E}_{\xi_i \sim \mathcal{P}_i}[l(x, \xi_i)]. \tag{1}$$

Here, the local objective function $f_i(x) : \mathbb{R}^n \mapsto \mathbb{R}$ represents the mathematical expectation of agent $i$'s loss function $l(x, \xi_i)$, where $\xi_i$ denotes the agent $i$'s data sample drawn from distribution $\mathcal{P}_i$.

In real-world applications, since the data distribution $\mathcal{P}_i$ is typically unknown to each agent $i$, it can only have access to a noisy estimate on the gradient of $f_i(x)$ (Pu & Nedić, 2021; Nazari et al., 2022; Kim et al., 2024). In other words, at each iteration $t$, each agent $i$ independently and identically samples $|\mathcal{B}|$ data points (also called a batch size of $|\mathcal{B}|$) from its local distribution $\mathcal{P}_i$ and computes a noisy gradient estimate $g_i^t(x) = \frac{1}{|\mathcal{B}|} \sum_{j=1}^{|\mathcal{B}|} \nabla l(x, \xi_{ij}^t)$, where $\xi_{ij}^t$ is the $j$th sampled data collected by agent $i$ at iteration $t$. Based on the gradient estimate $g_i^t(x)$ and communication with its neighbors, each agent $i$ performs distributed training. We make the following standard assumption about $f_i(x)$ and $g_i^t(x)$:

**Assumption 1.** *For any agent $i \in [m]$, its local objective function $f_i(x)$ is $\mu$-strongly convex and $L$-smooth. The gradient estimate $g_i^t(x)$ is unbiased with bounded variance $\sigma^2$, i.e., $\mathbb{E}[g_i^t(x)] = \nabla f_i(x)$ and $\mathbb{E}[\|g_i^t(x) - \nabla f_i(x)\|^2] \leq \frac{\sigma^2}{|\mathcal{B}|}$ hold for any $x \in \mathbb{R}^n$ and $t \geq 0$.*

In Assumption 1, the strong convexity of $f_i(x)$ is used to ensure linear convergence, which is commonly used in the existing literature (Ivkin et al., 2019; Hou et al., 2021; Akhavan et al., 2021; Wang et al., 2023; Yang & Ma, 2023; He et al., 2024; Er et al., 2024).

We describe the local interaction among agents using a weight matrix $W = \{w_{ij}\} \in \mathbb{R}^{m \times m}$, where $w_{ij} > 0$ if agent $i$ and agent $j$ can directly communicate with each other, and $w_{ij} = 0$ otherwise. The neighboring set of agent $i$ is defined as $\mathcal{N}_i = \{j \in [m]|w_{ij} > 0\}$, which includes itself. Furthermore, we make the following assumption on matrix $W$:

**Assumption 2.** *The matrix $W \in \mathbb{R}^{m \times m}$ is symmetric and satisfies $\mathbf{1}_m^\top W = \mathbf{1}_m^\top$, $W\mathbf{1}_m = \mathbf{1}_m$, and $\rho \triangleq \|W - \frac{\mathbf{1}_m \mathbf{1}_m^\top}{m}\| < 1$.*

Existing distributed optimization and learning approaches typically require the stepsize to be either fixed (Pu & Nedić, 2021; Koloskova et al., 2021; Nguyen et al., 2023; Song et al., 2024) or diminishing (Jakovetic et al., 2018; Dieuleveut & Patel, 2019; Li et al., 2024b; Lee et al., 2025). However, the use of a fixed stepsize often suffers from error/bias terms proportional to the stepsize (Yuan et al., 2016), which can cause pronounced overshoot or persistent oscillations near the global optimum, thereby compromising convergence stability in the later stages of the algorithm (as shown in our experimental results in Fig. 1). Although employing a diminishing stepsize can asymptotically eliminate such errors and ensure stable steady-state convergence, it often results in an undesirably low convergence speed, which is problematic for applications requiring fast convergence (Nedic & Ozdaglar, 2009; Jakovetic et al., 2018; Dieuleveut & Patel, 2019; Lee et al., 2025). Given these limitations, we aim to develop an adaptive stepsize approach for distributed stochastic optimization and learning, enabling each agent to adaptively adjust its stepsize during algorithm iterations to achieve both fast convergence and stable steady-state performance.

## 4 ALGORITHM DESIGN

In this section, we propose an adaptive stepsize approach for distributed stochastic optimization and learning that ensures both fast convergence and stable steady-state performance. The proposed approach is summarized in Algorithm 1, which is implemented in a fully distributed manner.

In Algorithm 1, Lines 3-7 execute a consensus-based gradient descent step for agent $i$. Lines 8, 11, and 14 update $y_{i,1}^{t+1}$ to track $\frac{1}{m} \sum_{i=1}^{m} g_i^t(x_i^{t+1})$, which serves to approximate the global gradient $\frac{1}{m} \sum_{i=1}^{m} \nabla f_i(x_i^{t+1})$. Lines 9, 12, and 14 update an auxiliary variable $y_{i,2}^t$ to track $\frac{1}{m} \sum_{i=1}^{m} g_i^t(x_i^t)$, which serves to approximate the global gradient $\frac{1}{m} \sum_{i=1}^{m} \nabla f_i(x_i^t)$. With this understanding, we let each agent $i$ locally estimate the curvature of the global objective function in Line 15. Based on this estimate, each agent $i$'s adaptive stepsize update rule is given in Line 15 and Line 16.

---

**Algorithm 1** Adaptive stepsize design for distributed stochastic optimization and learning (from agent $i$'s perspective)

---

1: **Input:** $x_i^0 \in \mathbb{R}^n$, $y_{i,1}^0 = g_i^{-1}(x_i^0) = g_i^0(x_i^0)$, $y_{i,2}^{-1} = g_i^{-1}(x_i^{-1}) = \mathbf{0}_n$, $\eta_i^0 > 0$, $\beta \in (1, 1.36)$, $r \in (0, 1)$, $M \in \mathbb{N}^+$, and $T \in \mathbb{N}^+$.
2: **for** $t = 0, 1, \ldots, T$ **do**
3: $\quad x_i^{t+1}(0) = x_i^t - \eta_i^t y_{i,1}^t$
4: $\quad$ **for** $q = 0, 1, \ldots, M - 1$ **do**
5: $\qquad x_i^{t+1}(q + 1) = \sum_{j \in \mathcal{N}_i} w_{ij} x_j^{t+1}(q)$
6: $\quad$ **end for**
7: $\quad x_i^{t+1} = x_i^{t+1}(M)$
8: $\quad y_{i,1}^{t+1}(0) = y_{i,1}^t + g_i^t(x_i^{t+1}) - g_i^{t-1}(x_i^t)$
9: $\quad y_{i,2}^t(0) = y_{i,2}^{t-1} + g_i^t(x_i^t) - g_i^{t-1}(x_i^{t-1})$
10: $\quad$ **for** $q = 0, 1, \ldots, M - 1$ **do**
11: $\qquad y_{i,1}^{t+1}(q + 1) = \sum_{j \in \mathcal{N}_i} w_{ij} y_{j,1}^{t+1}(q)$
12: $\qquad y_{i,2}^t(q + 1) = \sum_{j \in \mathcal{N}_i} w_{ij} y_{j,2}^t(q)$
13: $\quad$ **end for**
14: $\quad y_{i,1}^{t+1} = y_{i,1}^{t+1}(M) \quad$ and $\quad y_{i,2}^t = y_{i,2}^t(M)$
15: $\quad L_i^{t+1} = \frac{\|y_{i,1}^{t+1} - y_{i,2}^t\|}{\|x_i^{t+1} - x_i^t\|}$ if $x_i^{t+1} \neq x_i^t$; otherwise, $L_i^{t+1} = 1$
16: $\quad \eta_i^{t+1} = \min\left\{ \beta \eta_i^t, \frac{7\sqrt{r}}{10} \frac{\eta_i^t}{\sqrt{[m(\eta_i^t L_i^{t+1})^2 - 1]_+}} \right\}$
17: **end for**

---

The key enabler for us to ensure stable steady-state convergence is our meticulously designed stepsize update rule. More specifically, our stepsize update rule enables each agent to locally estimate the curvature of the global objective function. In this way, each agent's stepsize can be adapted to large values in the early stages of the algorithm, and to extremely small values near the global optimum (as shown in our experimental results in Figs. 1(d)-1(f) and Figs. 2(d)-2(f)). Therefore, our design avoids the slow convergence caused by small diminishing stepsizes used in, e.g., Jakovetic et al. (2018); Dieuleveut & Patel (2019); Li et al. (2024b); Lee et al. (2025) and eliminate the oscillations arising from fixed stepsizes in e.g., Pu & Nedić (2021); Koloskova et al. (2021); Nguyen et al. (2023); Song et al. (2024).

It is worth noting that our algorithm is fundamentally different from existing adaptive stepsize methods in e.g., Malitsky & Mishchenko (2019; 2024); Kim et al. (2024); Chen & Wang (2024), which explicitly require a centralized server to coordinate stepsize adjustment, which is infeasible in fully distributed settings in the absence of a centralized server. Furthermore, our design is also different from existing adaptive stepsize approaches for deterministic distributed optimization/learning in Carnevale et al. (2022); Xie et al. (2022); Saravanos et al. (2024), which typically require access to exact gradients of objective functions—such exact gradients are often unattainable in real-world applications where only noisy gradient estimates are available to each agent (Lian et al., 2017).

In Algorithm 1, we provide *optional* inner-consensus-loop iterations for $x_i^t$, $y_{i,1}^t$, and $y_{i,2}^t$. This design is intended to accelerate consensus among agents and improve the accuracy of global gradient tracking, thereby guaranteeing linear convergence (see Theorem 1 for details). In practical machine learning applications, the number of inner-consensus-loop iterations $M$ can be chosen as any positive integer. For example, we can simply select $M = 1$ (in which case Algorithm 1 reduces to a single-loop algorithm) to minimize the computational and communication costs of our algorithm. In fact, our experimental results in Fig. 3(c) show that the test accuracy of our algorithm remains comparable even with $M = 1$.

## 5 CONVERGENCE RESULTS

In this section, we prove that Algorithm 1 can ensure linear convergence with respect to the number of iterations $T$, and the convergence error decreases sublinearly with the batch size of sampled data. The results are summarized in Theorem 1, whose proof can be found in Appendix B.2.

**Theorem 1.** *Under Assumptions 1 and 2, for any $T \geq 0$ and batch size $|\mathcal{B}| > 0$, if the number of inner-consensus-loop iterations $M$ satisfies $M \geq M_0$ with detailed forms of $M_0$ given in equation 79 in Appendix B.2, the iterates $x_i^t$ generated by Algorithm 1 satisfy*

$$\mathbb{E}\left[\left\|x_i^T - x^*\right\|^2\right] \leq \mathcal{O}\left(\gamma^T\right) + \mathcal{O}\left(\frac{\sigma^2}{|\mathcal{B}|}\right), \tag{2}$$

*where the convergence rate $\gamma$ is given by $\gamma = \max\left\{1 - \frac{\mu}{4L}, \frac{91}{92}\right\}$.*

Theorem 1 proves that Algorithm 1 linearly converges to an optimal solution to problem 1 with the optimization error decreasing as the batch size of sampled data $|\mathcal{B}|$ increases. It is worth noting that the bound $\mathcal{O}\left(\frac{\sigma^2}{|\mathcal{B}|}\right)$ in Theorem 1, caused by finite batch size of sampled data, inherently exists in all stochastic optimization approaches with finite samples (Yuan et al., 2022; Sharma et al., 2023). Although variance reduction techniques (Reddi et al., 2016; Fang et al., 2018) and diminishing stepsize methods (Nedic & Ozdaglar, 2009) can be used to mitigate the influence of this term in distributed stochastic optimization and learning, their successful implementation heavily relies on the assumption of a fixed upper bound on the stepsizes, which is hard to satisfy when each agent's stepsize is dynamic and adaptive over iterations.

In Theorem 1, we consider a stochastic scenario in which each agent can only access to noisy gradient estimates (which are computed based on data sampled from an unknown data distribution $\mathcal{P}_i$). Next, we consider a deterministic scenario in which each agent can access to accurate gradients. The convergence result of Algorithm 1 in the deterministic scenario is summarized in the following Theorem 2, whose proof is given in Appendix B.3.

**Theorem 2.** *Under Assumptions 1 and 2, for any $T \geq 0$, if the number of inner-consensus-loop iterations $M$ satisfies $M \geq M_0$ with detailed forms of $M_0$ given in equation 79 of Appendix B.2, the iterates $x_i^t$ generated by Algorithm 1 with deterministic gradients satisfy*

$$\mathbb{E}\left[\left\|x_i^T - x^*\right\|^2\right] \leq \mathcal{O}\left(\gamma^T\right), \tag{3}$$

*where the convergence rate $\gamma$ is given by $\gamma = \max\left\{1 - \frac{\mu}{4L}, \frac{91}{92}\right\}$.*

Theorem 2 proves that when we consider distributed optimization and learning in a deterministic scenario, Algorithm 1 converges to an exact solution to problem in equation 1 with a linear convergence rate, which matches existing convergence results on adaptive stepsizes for centralized optimization and learning (Li & Orabona, 2019; Malitsky & Mishchenko, 2019; Kavis et al., 2022; Malitsky & Mishchenko, 2024). Moreover, this is also stronger than the convergence results achieved by existing distributed optimization methods with diminishing stepsizes (Jakovetic et al., 2018; Dieuleveut & Patel, 2019; Li et al., 2024b; Lee et al., 2025), which guarantee only sublinear convergence rates.

Furthermore, to give a more intuitive description of the computational complexity, we define an $\epsilon$-solution to problem in equation 1 as follows.

**Definition 1** (Lian et al. (2017)). *For some integer $T > 0$, if $\mathbb{E}[\|x_i^T - x^*\|^2] \leq \epsilon$ holds, then we say that the sequence $\{x_i^t\}$ can reach an $\epsilon$-solution to the problem in equation 1.*

Building on Theorem 1 and Theorem 2, we have the following corollary.

**Corollary 1.** *Under Assumptions 1 and 2, for any $\epsilon > 0$, Algorithm 1 with noisy gradient estimates requires at most $\mathcal{O}((2|\mathcal{B}| + 3M + 3)\log(\epsilon^{-1}))$ gradient evaluation to obtain an $\epsilon + \mathcal{O}(\frac{\sigma^2}{|\mathcal{B}|})$-solution, and Algorithm 1 with accurate gradients requires at most $\mathcal{O}((2M + 3)\log(\epsilon^{-1}))$ gradient evaluation to obtain an $\epsilon$-solution.*

In Corollary 1, the low bound on the number of inner-consensus-loop iterations $M$ in Algorithm 1 is a fixed constant, which is different from the existing distributed optimization results in, e.g., Berahas et al. (2019); Li et al. (2020) which have the inner-loop iteration number increasing with the outer-loop iterations, and hence have a higher computational complexity of the order of $\mathcal{O}((\log(\epsilon^{-1}))^2)$. Moreover, the computational complexity of our Algorithm 1 matches the adaptive stepsize results on centralized learning in, e.g., Malitsky & Mishchenko (2019; 2024) and the convergence results on distributed optimization in, e.g., Chen & Wang (2024); Kuruzov et al. (2024). This is also less than the convergence results in, e.g., Jakovetic et al. (2018); Dieuleveut & Patel (2019); Li et al. (2024b) with diminishing stepsizes which have a computation complexity of the order of $\mathcal{O}(\epsilon^{-1})$.

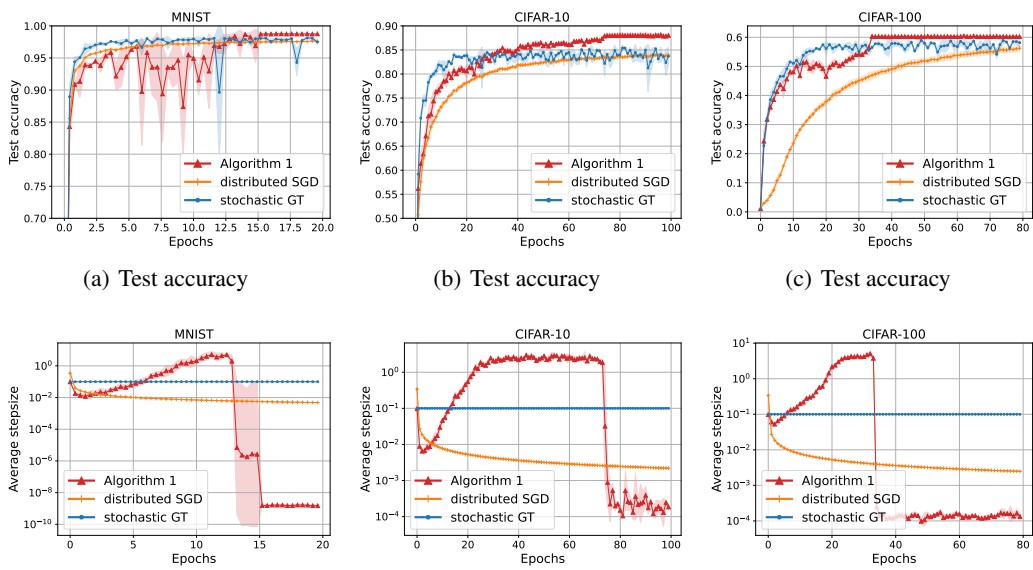

(a) Test accuracy      (b) Test accuracy      (c) Test accuracy

(d) Average stepsize across agents    (e) Average stepsize across agents    (f) Average stepsize across agents

Figure 1: Test-accuracy and average-stepsize (across five agents) evolutions of Algorithm 1, distributed SGD in Jakovetic et al. (2018), and stochastic GT in Pu & Nedić (2021) on the "MNIST", "CIFAR-10", and "CIFAR-100" datasets, respectively. The $95\%$ confidence intervals were computed from three independent runs with random seeds 42, 1010, and 2024.

## 6 EXPERIMENTS

In this section, we evaluate the performance of our proposed Algorithm 1 on image classification tasks using representative benchmark datasets, including the "MNIST" dataset (Deng, 2012), the "CIFAR-10" dataset (Krizhevsky et al., 2010), and "CIFAR-100" dataset (DeVries & Taylor, 2017). All these tasks involve nonsmooth and nonconvex objective functions, which are intended to show the effectiveness of our algorithm beyond the settings of strong convexity or smoothness. Due to the space limitations, we leave the experimental results on logistic regression with strongly convex and smooth loss functions to Appendix C.3. In all experiments, we considered five agents connected in a ring, where each agent communicates only with its two immediate neighbors. For the coupling matrix $W$, we set $w_{ii} = 0.4$ for all agent $i$, $w_{ij} = 0.3$ if agents $i$ and $j$ are neighbors, and $w_{ij} = 0$ otherwise. For each experiment, we considered heterogeneous data distribution, with each agent $i$ randomly sampling $40\%$ data points from the class $i$ and sampling $60\%$ data points from each remaining class. We evaluated the performance of our proposed algorithm through the following three cases: 1) we compared Algorithm 1 with existing distributed stochastic optimization/learning approaches, including distributed SGD in Jakovetic et al. (2018) with diminishing stepsize and the stochastic gradient-tracking (called stochastic GT) in Pu & Nedić (2021) with fixed stepsize; 2) we compared Algorithm 1 with existing adaptive stepsize approaches for centralized learning, including the well-known ADAM in Kingma (2014) and the adaptive SGD in Malitsky & Mishchenko (2024); and 3) to evaluate the effect of the coefficients $\beta$ and $r$ in the stepsize update rule (i.e., Line 16 in Algorithm 1) and the number of inner-consensus-loop iterations $M$ in Algorithm 1 on convergence accuracy, we test the convergence performance of Algorithm 1 under different $\beta$, $r$, and $M$, respectively. The detailed experimental settings are given in Appendix C.1 and Appendix C.2, and additional experimental results on comparison of Algorithm 1 and distributed ADAM in Nazari et al. (2022) are provided in Appendix C.3. The code for all experiments is available online[2].

**Comparison with existing distributed stochastic optimization approaches.** We trained convolutional neural networks (CNNs) with two, four, and five layers on the "MNIST", "CIFAR-10", and "CIFAR-100" datasets, respectively. We conducted training for 20 epochs on the "MNIST" dataset

---

[2]https://anonymous.4open.science/r/DASGD-71D1/README.md

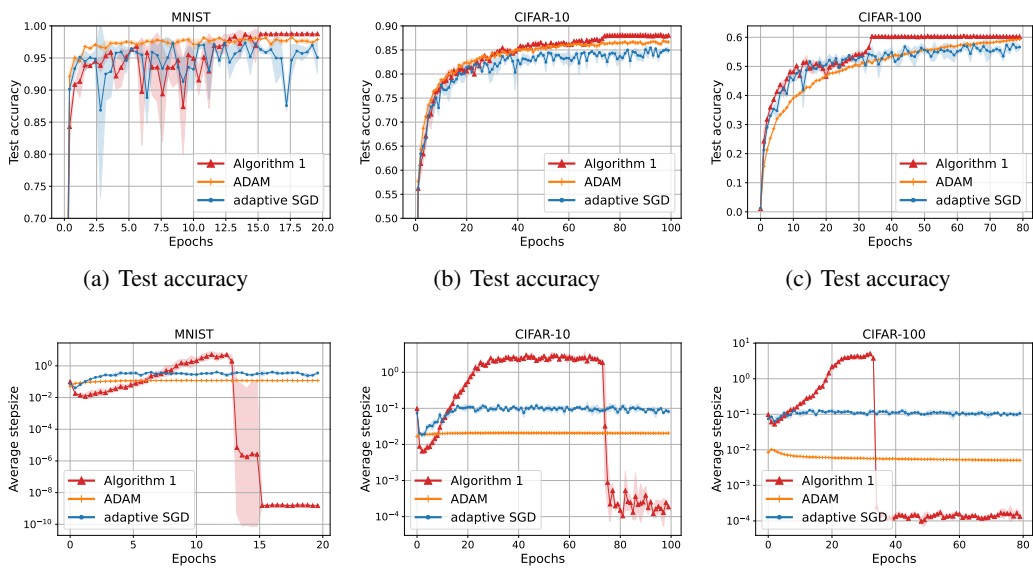

Figure 2: Test-accuracy and average-stepsize (across five agents) evolutions of Algorithm 1, ADAM in Kingma (2014), and adaptive SGD in Malitsky & Mishchenko (2024) on the "MNIST", "CIFAR-10", and "CIFAR-100" datasets, respectively. The 95% confidence intervals were computed from three independent runs with random seeds 42, 1010, and 2024.

and 80 epochs on the "CIFAR-10" and "CIFAR-100" datasets, using a batch size of 128. The step-size for distributed SGD was set as $\eta_i = \frac{0.1}{(t+1)^{0.5}}$ and for stochastic GT was set as $\eta_i = 0.1$. Both of them represent the best-performing stepsizes we could find in our comparison. In fact, during our tuning process, we obverse that setting $\eta = 0.01$ for stochastic GT results in overly slow convergence, whereas setting $\eta = 1$ leads to divergent behaviors. For Algorithm 1, we set the coefficients $\beta$ and $r$ in stepsize update rule as $\beta = 1.3$ and $r = 0.99$, and the number of inner-loop iterations as $M = 10$. (The test accuracies of Algorithm 1 under different $\beta$, $r$, and $M$ are provided in Figs. 3(a), 3(b), and 3(c), respectively.)

Fig. 1(a) to Fig. 1(c) show that our proposed Algorithm 1 achieves the highest test accuracy and a more stable steady-state convergence compared with distributed SGD in Jakovetic et al. (2018) and stochastic GT in Pu & Nedić (2021). The early-stage oscillations in test accuracy of Algorithm 1 are mainly attributable to the adaptive process of stepsize adjustments. Compared with distributed SGD with diminishing stepsizes, stochastic GT with a fixed stepsize achieves faster convergence, however, it suffers from larger steady-state oscillations. In contrast, our proposed algorithm eliminates steady-state oscillations, and hence, ensures fast convergence. This is achieved because our proposed adaptive stepsize rule allows each agent to take large stepsizes in the early stages of Algorithm 1 and extremely small stepsizes near the global optimum in the later stages, as shown in Fig. 1(d) to Fig. 1(f). These results further imply a clear stopping criterion for each agent in the implementation of our Algorithm 1. Specifically, we can preset a constant $\tau > 0$ (e.g., $\tau = 10^{-9}$ in the "MNIST" experiment) for all agents, and once an agent $i$'s stepsize $\eta_i^t$ falls below $\tau$, it can terminate training, which does not compromise the global learning accuracy.

**Comparison with existing adaptive stepsize approaches.** Since adaptive stepsize approaches are rarely reported in a fully distributed setting without a centralized server/aggregator, we compared the convergence performance of Algorithm 1 with that of existing adaptive stepsize approaches for centralized learning, including ADAM in Kingma (2014) and the adaptive SGD in Malitsky & Mishchenko (2019; 2024). This comparison is challenging because centralized methods can perform training directly on aggregated data, while our approach in Algorithm 1 operates in a fully distributed manner where each agent can only perform local computations and neighboring communication.

Fig. 2(a) to Fig. 2(c) show that Algorithm 1 has a higher test accuracy than both ADAM and adaptive SGD, even without the assistance of any centralized server/aggregator. This finding is noteworthy, as

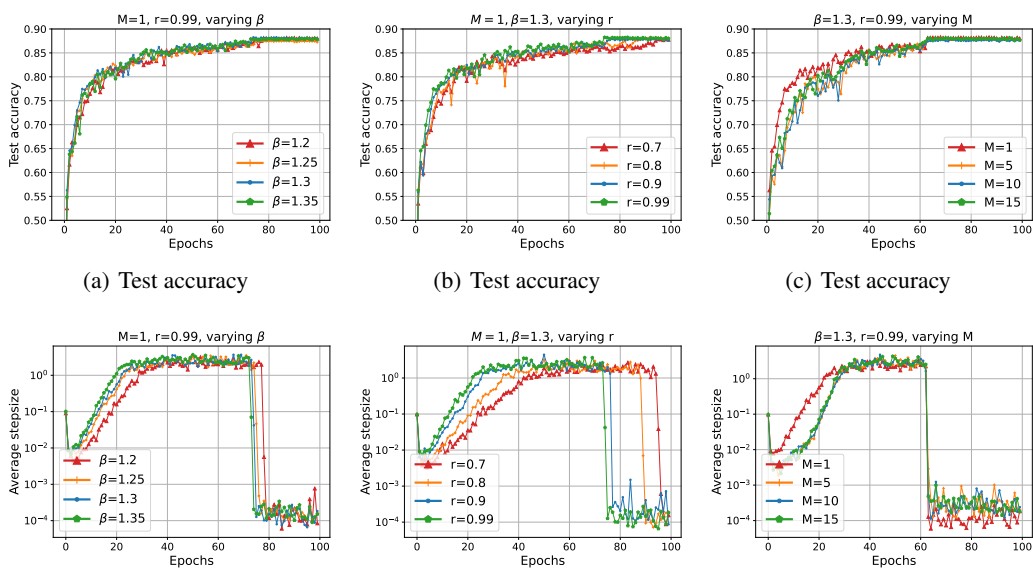

(a) Test accuracy        (b) Test accuracy        (c) Test accuracy

(d) Average stepsize across agents    (e) Average stepsize across agents    (f) Average stepsize across agents

Figure 3: Test-accuracy and average-stepsize (across five agents) evolutions of Algorithm 1 under different parameters $\beta$, $r$, and $M$, respectively, on the "CIFAR-10" dataset.

it empirically demonstrate that our fully distributed learning approach with heterogeneous adaptive stepsizes among agents can accelerate learning compared with centralized methods with a single adaptive stepsize. Furthermore, Fig. 2(d) to Fig. 2(f) once again confirm that our adaptive stepsize approach provides agents with large stepsizes in the early stages and small stepsizes in the convergence stages, thereby facilitating better performance than existing centralized counterparts.

**The effects of $\beta$, $r$, and $M$ on convergence accuracy.** We evaluate the test accuracies of Algorithm 1 under different coefficients $\beta$ and $r$ in the stepsize update rule (i.e., Line 16 in Algorithm 1) and the number of inner-loop iterations $M$ in Algorithm 1, respectively. We ran this experiment on the "CIFAR-10" dataset over 100 epochs, with a batch size of 64 and a random seed as 1010.

Fig. 3(a), Fig. 3(b), Fig. 3(d), and Fig. 3(e) imply that larger $\beta$ and $r$ lead to faster convergence and earlier stopping in Algorithm 1. This result is intuitively consistent, as large $\beta$ and $r$ contribute to larger stepsizes before convergence stages (as shown in Fig. 3(d) and Fig. 3(e)), which in turn leads to a higher convergence speed. Furthermore, Fig. 3(c) and Fig. 3(f) show that the number of inner-consensus-loop iterations $M$ has a negligible effect on convergence accuracy and the stopping criterion. Hence, in practical machine learning tasks, we can set $M = 1$ (so that Algorithm 1 reduces to a single-loop algorithm) to minimize the communication cost of Algorithm 1. In addition, the experimental results in Fig. 3 also suggest a default parameter configuration $(\beta, r, M) = (1.35, 0.99, 1)$ for Algorithm 1, which helps ease the tuning effort in real-world applications.

**The effect of network size $m$ on convergence accuracy.** We also evaluate the test accuracies of Algorithm 1 under different network sizes $m = 10$, $m = 15$, and $m = 20$, respectively. This experiment is conducted on the "CIFAR-10" dataset over 100 epochs with a batch size of 64 and a fixed random seed of 42. The remaining parameter settings are the same as those presented in the subsection "Comparison with existing distributed stochastic optimization approaches."

Fig. 4 shows that Algorithm 1 achieves higher test accuracy and more stable steady-state convergence than distributed SGD and stochastic GT, regardless of the network size $m$. Furthermore, we observe that a larger network size (i.e., a larger number of agents) leads to lower convergence accuracy under a fixed number of epochs. This is because increasing the network size reduces the number of training samples held by each agent. With a fixed batch size of 128, this reduction in local training samples decreases the number of iterations performed by each agent in each epoch, and consequently results in lower convergence accuracy within 100 epochs.

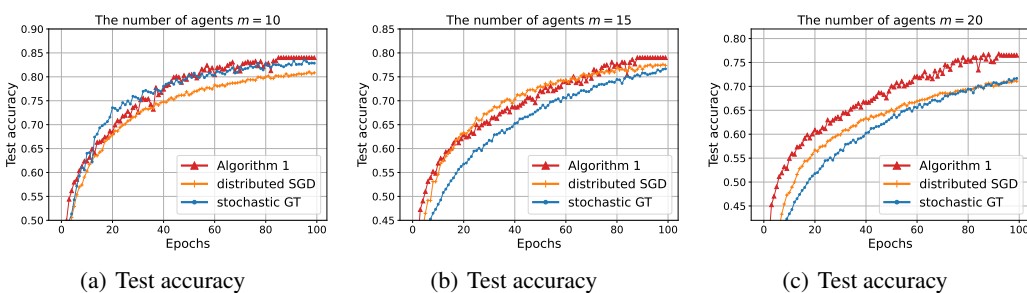

(a) Test accuracy     (b) Test accuracy     (c) Test accuracy

Figure 4: Test-accuracy evolutions of Algorithm 1, distributed SGD in Jakovetic et al. (2018), and stochastic GT in Pu & Nedić (2021) under different network sizes $m = 10$, $m = 15$, and $m = 20$, respectively, on the "CIFAR-10" dataset.

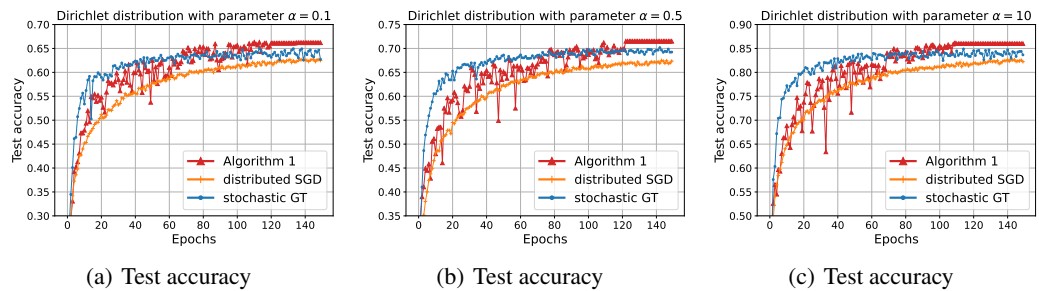

(a) Test accuracy     (b) Test accuracy     (c) Test accuracy

Figure 5: Test-accuracy evolutions of Algorithm 1, distributed SGD in Jakovetic et al. (2018), and stochastic GT in Pu & Nedić (2021) under different Dirichlet-distribution parameter $\alpha = 0.1$, $\alpha = 0.5$, and $\alpha = 10$, respectively, on the "CIFAR-10" dataset.

**The effect of data heterogeneity across agents on convergence accuracy.** Furthermore, to evaluate the convergence performance of our algorithm under different data distributions across agents, we conduct experiments on the "CIFAR-10" dataset using the Dirichlet partitioning scheme with parameters $\alpha = 0.1$, $\alpha = 0.5$, and $\alpha = 10$ (note that a smaller $\alpha$ corresponds to a higher level of data heterogeneity among agents). The remaining experimental settings follow those presented in previous subsection "The effect of network size $m$ on convergence accuracy."

Fig. 5 shows that Algorithm 1 maintains higher test accuracy and more stable steady-state convergence than both distributed SGD and stochastic GT under all levels of data heterogeneity among agents. In addition, it can be seen that a larger $\alpha$ (i.e., a lower level of data heterogeneity across agents) leads to higher convergence accuracy.

## 7 CONCLUSION

In this paper, we have proposed an adaptive stepsize approach for distributed stochastic optimization and learning without the assistance of any centralized server/aggregator or the need for accurate gradients. This is nontrivial, because existing adaptive stepsize approaches either rely on a centralized server to coordinate stepsizes among agents, or are limited to deterministic scenarios where agents have access to accurate gradients of the objective functions. Moreover, our approach can eliminate steady-state oscillations, and hence, ensures fast convergence. This stands in stark contrast to most existing adaptive stepsize approaches that often incur steady-state oscillations near the global optimal solution, and thereby preventing the algorithm from achieving stable convergence accuracy. In addition, we have systematically characterized the convergence rates of our algorithm for both stochastic and deterministic distributed optimization, and quantified the computational complexities for gradient evaluations on both cases. Experimental results on image classifications using three benchmark datasets confirm the advantages of the proposed approach over existing counterparts.

**Ethics statement.** All authors declare no conflicts of interest and no ethical issues in this work.

**Reproducibility statement.** All authors confirm the reproducibility of both the theoretical and experimental results. The code for all experiments is available online at `https://anonymous.4open.science/r/DASGD-71D1/README.md`. Detailed descriptions of the experimental settings and implementation details are provided in the main text and Appendix. Theoretical assumptions are clearly stated, and complete proofs of all results are included in the Appendix.

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

APPENDIX

## A  NOTATIONS

For the sake of notational simplicity, we introduce some additional notations. We use $\mathbb{R}^+$ to denote the set of positive real numbers and use $\mathbf{X}^t \triangleq \mathrm{col}(x_1^t, x_2^t, \cdots, x_m^t) \in \mathbb{R}^{mn}$ to denote the stacked model parameters of all agents. We also use $\otimes$ to denote the Kronecker product. We use $\overline{x}^t(q)$ to denote the average of all agents' model parameters at the $q$th inner iteration of the $t$th outer iteration. We define $\mathcal{F}_t \triangleq \{\xi_{i,s} | i = 1, \cdots, m \text{ and } s = 0, \cdots, t\}$, where $\xi_{i,t}$ represents the data point sampled by agent $i$ at the $t$th iteration. For further notational simplicity, we define $\overline{x}^t = \frac{1}{m}\sum_{i=1}^m x_i^t$, $\overline{\eta^t y_1^t} = \frac{1}{m}\sum_{i=1}^m \eta_i^t y_{1,i}^t$, $\eta_{\max}^t = \max_{i \in [m]} \eta_i^t$, $\eta_{\max} = \max_{t \in \mathbb{N}} \eta_{\max}^t$, $\overline{\eta}^t = \frac{1}{m}\sum_{i=1}^m \eta_i^t$, $\overline{y}_1^t = \frac{1}{m}\sum_{i=1}^m y_{1,i}^t$, $\overline{y}_2^t = \frac{1}{m}\sum_{i=1}^m y_{2,i}^t$, $\hat{x}_i^t = x_i^t - \overline{x}^t$, $\hat{y}_{1,i}^t = y_{1,i}^t - \overline{y}_1^t$, and $\hat{y}_{2,i}^t = y_{2,i}^t - \overline{y}_2^t$.

## B  RESULTS OF ALGORITHM 1

### B.1  TECHNICAL LEMMAS

We introduce the following three lemmas to characterize the consensus errors of Algorithm 1.

**Lemma 1.** *Under Assumptions 1 and 2, the following inequality holds for Algorithm 1:*

$$\mathbb{E}[\|\overline{x}^{t+1} - \overline{x}^t\|^2] \leq \frac{45}{46}\mathbb{E}[\|\overline{x}^t - \overline{x}^{t-1}\|^2] - \mathbb{E}[\|\overline{x}^{t+1} - \overline{x}^t\|^2]$$
$$+ \frac{125}{31\beta}\mathbb{E}[\overline{\eta}^t(f(\overline{x}^{t-1}) - f(\overline{x}^t))] + 2\delta_2^t, \tag{4}$$

*where the constant $\delta_3^t$ is given by*

$$\delta_2^t = \frac{b_{\hat{x},1}}{m}\sum_{i=1}^m \mathbb{E}[\|\hat{x}_i^t\|^2] + \frac{b_{\hat{x},2}}{m}\sum_{i=1}^m \mathbb{E}[\|\hat{x}_i^{t-1}\|^2] + \frac{b_{\hat{x},2}}{m}\sum_{i=1}^m \mathbb{E}[\|\hat{x}_i^t(0)\|^2] + \frac{b_{\hat{y},1}}{m}\sum_{i=1}^m \mathbb{E}[\|\hat{y}_{1,i}^t\|^2]$$
$$+ \frac{b_{\hat{y},2}}{m}\sum_{i=1}^m \mathbb{E}[\|\hat{y}_{1,i}^{t-1}\|^2] + \frac{b_{\hat{y},3}}{m}\sum_{i=1}^m \mathbb{E}[\|\hat{y}_{2,i}^{t-1}\|^2] + \frac{b_\sigma \sigma^2}{|\mathcal{B}|}, \tag{5}$$

*where $b_{\hat{x},1} = \frac{125}{124}\left(\frac{L^2\eta_{\max}}{a_8} + \frac{2\beta^2}{a_7}\right) + 4\beta^2 + 2$, $b_{\hat{x},2} = 2(1 - a_4)(1 - a_5)\left(1 - \frac{1}{a_6}\right)\beta^2 + 4\beta^2 + 2$, $b_{\hat{y},1} = 4\eta_{\max}^2\left(1 + \frac{1}{a_2}\right)$, $b_{\hat{y},2} = 2(1 - a_5)\left(1 - \frac{1}{a_6}\right)\eta_{\max}^2$. $b_{\hat{y},3} = 4\eta_{\max}^2\left(1 + \frac{1}{a_2}\right)$, and $b_\sigma = 4\left(1 - \frac{1}{a_4}\right)\eta_{\max}^2$.*

*Proof.* According to Line 3 in Algorithm 1, we have

$$\overline{x}^{t+1} = \overline{x}^{t+1}(0) = \overline{x}^t - \overline{\eta^t y_1^t}, \tag{6}$$

with $\overline{x}^t = \frac{1}{m}\sum_{i=1}^m x_i^t$ and $\overline{\eta^t y_1^t} = \frac{1}{m}\sum_{i=1}^m \eta_i^t y_{1,i}^t$. Since $\overline{\eta}^t\overline{y}_1^t = \frac{1}{m^2}\sum_{i=1}^m \eta_i^t \sum_{j=1}^m y_{1,j}^t$ holds, we obtain the following inequality:

$$
\begin{aligned}
\mathbb{E}[\|\overline{x}^{t+1} - \overline{x}^t\|^2 | \mathcal{F}_t] &= \|\overline{\eta^t y_1^t}\|^2 = \|\overline{\eta^t y_1^t} - \overline{\eta}^t\overline{y}_1^t + \overline{\eta}^t\overline{y}_1^t\|^2 \\
&\leq \left(1 + \frac{1}{a_1}\right)\|\overline{\eta^t y_1^t} - \overline{\eta}^t\overline{y}_1^t\|^2 + (1+a_1)\|\overline{\eta}^t\overline{y}_1^t\|^2 \\
&= \left(1 + \frac{1}{a_1}\right)\|\frac{1}{m}\sum_{i=1}^m(\eta_i^t y_{1,i}^t - \eta_i^t\overline{y}_1^t)\|^2 + (1+a_1)\|\overline{\eta}^t\overline{y}_1^t\|^2 \\
&\leq \left(1 + \frac{1}{a_1}\right)\frac{1}{m}\sum_{i=1}^m(\eta_i^t)^2\|y_{1,i}^t - \overline{y}_1^t\|^2 + (1+a_1)\|\overline{\eta}^t\overline{y}_1^t\|^2 \\
&\leq \left(1 + \frac{1}{a_1}\right)\frac{1}{m}\sum_{i=1}^m \eta_{\max}^2\|\hat{y}_{1,i}^t\|^2 + (1+a_1)\|\overline{\eta}^t\overline{y}_1^t\|^2,
\end{aligned}
\tag{7}
$$

where $\mathcal{F}_t = \{\xi_{i,s} | i = 1, \ldots, N; \; s = 0, \ldots, t\}$ with $\xi_{i,t}$ denoting the data point sampled by agent $i$ at iteration $t$. Here, we have used the inequality $\|a+b\|^2 \leq (1+\frac{1}{\alpha})\|a\|^2 + (1+\alpha)\|b\|^2$ for any $\alpha > 0$ and $a,\, b \in \mathbb{R}^n$ in the first inequality and the inequality $\|\frac{1}{m}\sum_{i=1}^m a_i\|^2 \leq \frac{1}{m}\sum_{i=1}^m\|a_i\|^2$ for any $a_i \in \mathbb{R}^n$, $i = 1, \cdots, m$ in the second inequality. By choosing $a_1 \in (0, \frac{1}{124})$ and applying the relation $\|a\|^2 = \|a-b\|^2 - \|b\|^2 + 2\langle a, b\rangle$ to equation 7, the term $\|\overline{\eta}^t\overline{y}_1^t\|$ can be bounded by

$$
\|\overline{\eta}^t\overline{y}_1^t\|^2 = \|\overline{\eta}^t(\overline{y}_1^t - \overline{y}_2^{t-1})\|^2 - \|\overline{\eta}^t\overline{y}_2^{t-1}\|^2 + 2\langle\overline{\eta}^t\overline{y}_1^t, \overline{\eta}^t\overline{y}_2^{t-1}\rangle.
\tag{8}
$$

The first term on the right-hand side of equation 8 satisfies

$$
\begin{aligned}
\|\overline{\eta}^t(\overline{y}_1^t - \overline{y}_2^{t-1})\|^2 &= \left\|\frac{1}{m}\sum_{i=1}^m \eta_i^t(\overline{y}_1^t - \overline{y}_2^{t-1})\right\|^2 = \left\|\frac{1}{m}\sum_{i=1}^m \eta_i^t(y_{1,i}^t - y_{2,i}^{t-1}) - \eta_i^t(\hat{y}_{1,i}^t - \hat{y}_{2,i}^{t-1})\right\|^2 \\
&\leq \left\|\frac{1}{m}\sum_{i=1}^m(1+a_2)\eta_i^t(y_{1,i}^t - y_{2,i}^{t-1})\right\|^2 + \frac{1}{m}\sum_{i=1}^m\left(1 + \frac{1}{a_2}\right)\|\eta_i^t(\hat{y}_{1,i}^t - \hat{y}_{2,i}^{t-1})\|^2, \\
&\leq \left\|\frac{1}{m}\sum_{i=1}^m(1+a_2)\eta_i^t L_i^t(x_i^t - x_i^{t-1})\right\|^2 + \frac{1}{m}\sum_{i=1}^m\left(1 + \frac{1}{a_2}\right)\|\eta_i^t(\hat{y}_{1,i}^t - \hat{y}_{2,i}^{t-1})\|^2,
\end{aligned}
\tag{9}
$$

for any $a_2 \in \mathbb{R}^+$.

Consider the second term on the right-hand side of equation 9.

$$
\mathbb{E}\left[\frac{1}{m}\sum_{i=1}^m\left(1 + \frac{1}{a_2}\right)\|\eta_i^t(\hat{y}_{1,i}^t - \hat{y}_{2,i}^{t-1})\|^2\right] \leq \frac{2}{m}\sum_{i=1}^m \eta_{\max}^2\left(1 + \frac{1}{a_2}\right)\mathbb{E}\left[\|\hat{y}_{1,i}^t\|^2 + \|\hat{y}_{2,i}^{t-1}\|^2\right].
\tag{10}
$$

Substituting equation 10 into equation 9 yields

$$
\begin{aligned}
\mathbb{E}\left[\|\overline{\eta}^t(\overline{y}_1^t - \overline{y}_2^{t-1})\|^2\right] &\leq \mathbb{E}\left[\left\|\frac{1}{m}\sum_{i=1}^m(1+a_2)\eta_i^t L_i^t(x_i^t - x_i^{t-1})\right\|^2\right] \\
&+ \frac{2}{m}\sum_{i=1}^m \eta_{\max}^2\left(1 + \frac{1}{a_2}\right)\mathbb{E}\left[\|\hat{y}_{1,i}^t\|^2 + \|\hat{y}_{2,i}^{t-1}\|^2\right].
\end{aligned}
\tag{11}
$$

We proceed to estimate a lower bound on the second term on the right-hand side of equation 8.

$$
\begin{aligned}
\|\overline{\eta}^t\overline{y}_2^{t-1}\|^2 &= (\overline{\eta}^t)^2\|\overline{y}_2^{t-1}\|^2 = (\overline{\eta}^t)^2\|\overline{y}_2^{t-1} - \overline{y}_1^{t-1} + \overline{y}_1^{t-1}\|^2 \\
&\geq (1-a_3)(\overline{\eta}^t)^2\|\overline{y}_1^{t-1}\|^2 + \left(1 - \frac{1}{a_3}\right)(\overline{\eta}^t)^2\|\overline{y}_2^{t-1} - \overline{y}_1^{t-1}\|^2,
\end{aligned}
\tag{12}
$$

for any $a_3 \in (0, 1)$, where in the derivation we have used the inequality $\|a+b\|^2 \geq (1-\frac{1}{\alpha})\|a\|^2 + (1-\alpha)\|b\|^2$, for any $a \in (0, 1)$.

Since the relationship $1 - \frac{1}{a_4} < 0$ holds, the second term on the right-hand side of equation 12 satisfies

$$
\begin{aligned}
&\left(1 - \frac{1}{a_4}\right) \mathbb{E}\left[(\overline{\eta}^t)^2 \|\overline{y}_2^{t-1} - \overline{y}_1^{t-1}\|^2\right] \\
&\geq \left(1 - \frac{1}{a_4}\right)(\eta_{\max})^2 \frac{1}{m}\sum_{i=1}^m \mathbb{E}\left[\|(g_i^{t-1}(x_i^{t-1}) - g_i^{t-2}(x_i^{t-1}))\|^2\right] \geq 2\left(1 - \frac{1}{a_4}\right)\frac{\eta_{\max}^2 \sigma^2}{|\mathcal{B}|}.
\end{aligned}
\tag{13}
$$

By using the inequality $\sum_{i=1}^n a_i^2 \leq \left(\sum_{i=1}^n a_i\right)^2 \leq n\sum_{i=1}^n a_i^2$ for any nonnegative constants $a_1, \ldots, a_n$, the first term on the right-hand side of equation 12 satisfies

$$
\begin{aligned}
\mathbb{E}\left[(\overline{\eta}^t)^2\|\overline{y}_1^{t-1}\|^2\right] &= \mathbb{E}\left[\left(\|\overline{\eta}^t \overline{y}_1^{t-1}\|\right)^2\right] = \mathbb{E}\left[\left(\frac{1}{m}\sum_{i=1}^m \|\eta_i^t \overline{y}_1^{t-1}\|\right)^2\right] \\
&= \mathbb{E}\left[\left(\frac{1}{m}\sum_{i=1}^m \eta_i^t \|y_{1,i}^{t-1} - \hat{y}_{1,i}^{t-1}\|\right)^2\right] \\
&\geq (1 - a_5)\mathbb{E}\left[\left\|\frac{1}{m}\sum_{i=1}^m \eta_i^t y_{1,i}^{t-1}\right\|^2\right] + \left(1 - \frac{1}{a_5}\right)\frac{\eta_{\max}^2}{m}\mathbb{E}\left[\|\hat{y}_{1,i}^{t-1}\|^2\right],
\end{aligned}
\tag{14}
$$

holds for any $a_5 \in (0,1)$.

We estimate a lower bound on the first term on the right-hand side of equation 14 as follows:

$$
\begin{aligned}
\mathbb{E}\left[\left\|\frac{1}{m}\sum_{i=1}^m \eta_i^t y_{1,i}^{t-1}\right\|^2\right] &= \left[\left(\frac{1}{m}\sum_{i=1}^m \frac{\eta_i^t}{\eta_i^{t-1}}\|x_i^t(0) - x_i^{t-1}\|\right)^2\right] \\
&\geq (1 - a_6)\mathbb{E}\left[\left(\frac{1}{m}\sum_{i=1}^m \frac{\eta_i^t}{\eta_i^{t-1}}\|\overline{x}^t - \overline{x}^{t-1}\|\right)^2\right] \\
&\quad + \left(1 - \frac{1}{a_6}\right)\sum_{i=1}^m \mathbb{E}\left[\left(\frac{1}{m}\sum_{i=1}^m \frac{\eta_i^t}{\eta_i^{t-1}}\|\hat{x}_i^t(0) - \hat{x}_i^{t-1}\|\right)^2\right],
\end{aligned}
\tag{15}
$$

for any $a_6 \in (0,1)$.

By the inequality $\rho^{2M}\sum_{i=1}^m \|\hat{x}_i^t(0)\|^2 \geq \sum_{i=1}^m \|\hat{x}_i^t\|^2$, which will be proved in the subsequent Lemma 3, we have

$$
\sum_{i=1}^m \mathbb{E}\left[\left(\frac{1}{m}\sum_{i=1}^m \frac{\eta_i^t}{\eta_i^{t-1}}\|\hat{x}_i^t(0) - \hat{x}_i^{t-1}\|\right)^2\right] \leq \beta^2 \frac{1}{m}\sum_{i=1}^m \mathbb{E}\left[\|\hat{x}_i^t(0)\|^2 + \rho^{2M}\|\hat{x}_i^{t-1}(0)\|^2\right].
\tag{16}
$$

Finally, using inequalities equation 12 -equation 16, we arrive at

$$
\begin{aligned}
\mathbb{E}[\|\overline{\eta}^t \overline{y}_2^{t-1}\|^2] &\geq (1 - a_4)(1 - a_5)(1 - a_6)\mathbb{E}\left[\left(\frac{1}{m}\sum_{i=1}^m \frac{\eta_i^t}{\eta_i^{t-1}}\|\overline{x}^t - \overline{x}^{t-1}\|\right)^2\right] \\
&\quad + 2\left(1 - \frac{1}{a_4}\right)\frac{\eta_{\max}^2 \sigma^2}{|\mathcal{B}|} + (1 - a_4)\left(1 - \frac{1}{a_5}\right)\frac{\eta_{\max}^2}{m}\sum_{i=1}^m \mathbb{E}\left[\|\hat{y}_{1,i}^{t-1}\|^2\right] \\
&\quad + (1 - a_4)(1 - a_5)\left(1 - \frac{1}{a_6}\right)\beta^2 \frac{1}{m}\sum_{i=1}^m \mathbb{E}\left[\|\hat{x}_i^t(0)\|^2 + \rho^{2M}\|\hat{x}_i^{t-1}(0)\|^2\right].
\end{aligned}
\tag{17}
$$

Next, we estimate an upper bound for the last term on the right-hand side of equation 8 as follows:

$$2\mathbb{E}\left[\langle \overline{\eta}^t \overline{y}_1^t, \overline{\eta}^t \overline{y}_2^{t-1} \rangle\right] = \frac{2}{m} \sum_{i=1}^{m} \mathbb{E}\left[\langle \overline{\eta}^t \overline{y}_1^t, \eta_i^t(\overline{y}_2^{t-1} - \overline{y}_1^{t-1} + \overline{y}_1^{t-1})\rangle\right]$$

$$= \frac{2}{m} \sum_{i=1}^{m} \mathbb{E}\left[\langle \overline{\eta}^t \overline{y}_1^t, \eta_i^t y_{1,i}^{t-1}\rangle\right] - \frac{2}{m} \sum_{i=1}^{m} \mathbb{E}\left[\langle \overline{\eta}^t \overline{y}_1^t, \eta_i^t \hat{y}_{1,i}^{t-1}\rangle\right] + \frac{2}{m} \sum_{i=1}^{m} \mathbb{E}\left[\langle \overline{\eta}^t \overline{y}_1^t, \eta_i^t(\overline{y}_2^{t-1} - \overline{y}_1^{t-1})\rangle\right]$$

$$= \frac{2}{m} \sum_{i=1}^{m} \mathbb{E}\left[\left\langle \overline{\eta}^t \overline{y}_1^t, \frac{\eta_i^t}{\eta_i^{t-1}}(x_i^{t-1} - x_i^t(0))\right\rangle\right] - \frac{2}{m} \sum_{i=1}^{m} \mathbb{E}\left[\langle \overline{\eta}^t \overline{y}_1^t, \eta_i^t \hat{y}_{1,i}^{t-1}\rangle\right]$$

$$+ \frac{2}{m} \sum_{i=1}^{m} \mathbb{E}\left[\langle \overline{\eta}^t \overline{y}_1^t, \eta_i^t(\overline{y}_2^{t-1} - \overline{y}_1^{t-1})\rangle\right]$$

$$\leq \frac{2}{m} \sum_{i=1}^{m} \mathbb{E}\left[\left\langle \overline{\eta}^t \overline{y}_1^t, \frac{\eta_i^t}{\eta_i^{t-1}}(x_i^{t-1} - x_i^t(0))\right\rangle\right] + 2a_7 \mathbb{E}\left[\|\overline{\eta}^t \overline{y}_1^t\|^2\right] + \frac{1}{m} \sum_{i=1}^{m} \mathbb{E}\left[\frac{\eta_{\max}^2}{a_7}\|\hat{y}_{1,i}^{t-1}\|^2\right]$$

$$+ \mathbb{E}\left[\frac{\eta_{\max}^2}{a_7}\|\overline{y}_2^{t-1} - \overline{y}_1^{t-1}\|^2\right], \tag{18}$$

with $a_7 = \frac{1-124a_1}{250} > 0$, where we have used the inequality $2\langle a, b\rangle \leq \frac{1}{\alpha}\|a\|^2 + \alpha\|b\|^2$ in the last inequality.

Next, we need to transform the first term on the right-hand side of equation 18.

$$2\mathbb{E}\left[\left\langle \overline{\eta}^t \overline{y}_1^t, \frac{\eta_i^t}{\eta_i^{t-1}}(x_i^{t-1} - x_i^t)\right\rangle\right]$$

$$= 2\mathbb{E}\left[\left\langle \overline{\eta}^t \overline{y}_1^t, \frac{\eta_i^t}{\eta_i^{t-1}}(\overline{x}^{t-1} - \overline{x}^t)\right\rangle\right] + 2\mathbb{E}\left[\left\langle \overline{\eta}^t \overline{y}_1^t, \frac{\eta_i^t}{\eta_i^{t-1}}(\hat{x}_i^{t-1} - \hat{x}_i^t)\right\rangle\right]$$

$$\leq \frac{2}{m} \sum_{j=1}^{m} \mathbb{E}\left[\left\langle \eta_j^t \overline{y}_1^t, \frac{\eta_i^t}{\eta_i^{t-1}}(\overline{x}^{t-1} - \overline{x}^t)\right\rangle\right] + a_7 \mathbb{E}\left[\|\overline{\eta}^t \overline{y}_1^t\|^2\right] + \frac{2\beta^2}{a_7}\mathbb{E}\left[\|\hat{x}_i^{t-1}\|^2\right] + \frac{2\beta^2}{a_7}\mathbb{E}\left[\|\hat{x}_i^t\|^2\right]. \tag{19}$$

Then, we estimate an upper bound on the first term on the right-hand side of equation 19 as follows:

$$\mathbb{E}\left[\left\langle \eta_j^t \overline{y}_1^t, \frac{\eta_i^t}{\eta_i^{t-1}}(\overline{x}^{t-1} - \overline{x}^t)\right\rangle\right] = \mathbb{E}\left[\eta_j^t \left\langle \overline{y}_1^t - \nabla f(\overline{x}^t), \frac{\eta_i^t}{\eta_i^{t-1}}(\overline{x}^{t-1} - \overline{x}^t)\right\rangle\right]$$

$$+ \mathbb{E}\left[\eta_j^t \left\langle \nabla f(\overline{x}^t), \frac{\eta_i^t}{\eta_i^{t-1}}(\overline{x}^{t-1} - \overline{x}^t)\right\rangle\right]$$

$$\leq \mathbb{E}\left[\eta_j^t \left\langle \overline{y}_1^t - \nabla f(\overline{x}^t), \frac{\eta_i^t}{\eta_i^{t-1}}(\overline{x}^{t-1} - \overline{x}^t)\right\rangle\right] + \frac{1}{\beta}\mathbb{E}\left[\eta_j^t(f(\overline{x}^{t-1}) - f(\overline{x}^t))\right], \tag{20}$$

with $\nabla f(\overline{x}^t) = \frac{1}{m}\sum_{i=1}^{m} \nabla f_i(\overline{x}^t)$, where we have used the convexity of the function $f(x)$ and the relationship $\frac{\eta_i^t}{\eta_i^{t-1}} \leq \frac{1}{\beta}$ for any given $t$ in the last inequality.

The first term on the right-hand side of equation 20 can be bounded by

$$\frac{1}{m} \sum_{i=1}^{m} \mathbb{E}\left[\left\langle \overline{y}_1^t - \nabla f(\overline{x}^t), \frac{\eta_i^t}{\eta_i^{t-1}}(\overline{x}^{t-1} - \overline{x}^t)\right\rangle\right]$$

$$\leq \frac{1}{2a_8}\mathbb{E}\left[\|\overline{y}_1^t - \nabla f(\overline{x}^t)\|^2\right] + \frac{a_8}{2}\mathbb{E}\left[\left(\frac{1}{m}\sum_{i=1}^{m}\frac{\eta_i^t}{\eta_i^{t-1}}\right)^2 \|\overline{x}^{t-1} - \overline{x}^t\|^2\right], \tag{21}$$

Since the random variables $g_i^{t-1}(x_i^t) - \nabla f_i(x_i^t)$, $i = 1, \ldots, m$, are mutually independent, and $\mathbb{E}\left[g_i^{t-1}(x_i^t) - \nabla f_i(x_i^t)\right] = 0$, we have $\mathbb{E}\left[\left\|\frac{1}{m}\sum_{j=1}^{m}\left(g_i^{t-1}(x_i^t) - \nabla f_i(x_i^t)\right)\right\|^2\right] =$

$\frac{1}{m^2} \sum_{i=1}^m \mathbb{E} \left[ \left\| g_i^{t-1}(x_i^t) - \nabla f_i(x_i^t) \right\|^2 \right] = \frac{\sigma^2}{|\mathcal{B}|m}$. Using the $L$-smoothness of $f_j$, the relationship $\overline{y}_1^t = \frac{1}{m} \sum_{j=1}^m g_j^{t-1}(x_j^t)$, and the inequality $\left\| \frac{1}{m} \sum_{i=1}^m a_i \right\|^2 \leq \frac{1}{m} \sum_{i=1}^m \|a_i\|^2$ for any $a_1, \cdots, a_m \in \mathbb{R}^n$, we have

$$
\begin{aligned}
&\mathbb{E} \left[ \|\overline{y}_1^t - \nabla f(\overline{x}^t)\|^2 \right] \\
&= \mathbb{E} \left[ \left\| \frac{1}{m} \sum_{i=1}^m g_i^{t-1}(x_i^t) - \frac{1}{m} \sum_{i=1}^m \nabla f_i(\overline{x}^t) \right\|^2 \right] \\
&\leq 2\mathbb{E} \left[ \left\| \frac{1}{m} \sum_{i=1}^m \left( g_i^{t-1}(x_i^t) - \nabla f_i(x_i^t) \right) \right\|^2 \right] + 2\mathbb{E} \left[ \left\| \frac{1}{m} \sum_{i=1}^m \left( \nabla f_i(\bar{x}^t) - \nabla f_i(x_i^t) \right) \right\|^2 \right] \quad (22) \\
&\leq \frac{2}{m^2} \sum_{j=1}^m \mathbb{E} \left[ \left\| g_i^{t-1}(x_i^t) - \nabla f_i(x_i^t) \right\|^2 \right] + \frac{2}{m} \sum_{i=1}^m \mathbb{E} \left[ \left\| \nabla f_i(\bar{x}^t) - \nabla f_i(x_i^t) \right\|^2 \right] \\
&\leq \frac{2L^2}{m} \sum_{i=1}^m \mathbb{E} \left[ \|\hat{x}_i^t\|^2 \right] + \frac{2\sigma^2}{|\mathcal{B}|m}.
\end{aligned}
$$

Substituting equation 22 into equation 21 yields

$$
\begin{aligned}
&\frac{1}{m} \sum_{i=1}^m \mathbb{E} \left[ \left\langle \overline{y}_1^t - \nabla f(\overline{x}^t), \frac{\eta_i^t}{\eta_i^{t-1}} (\overline{x}^{t-1} - \overline{x}^t) \right\rangle \right] \\
&\leq \frac{L^2}{ma_8} \sum_{i=1}^m \mathbb{E} \left[ \|\hat{x}_i^t\|^2 \right] + \frac{a_8}{2} \mathbb{E} \left[ \left( \frac{1}{m} \sum_{i=1}^m \frac{\eta_i^t}{\eta_i^{t-1}} \right)^2 \|\overline{x}^{t-1} - \overline{x}^t\|^2 \right] + \frac{\sigma^2}{|\mathcal{B}|ma_8}, \quad (23)
\end{aligned}
$$

Combining equation 18 and equation 23, we obtain the following inequality:

$$
\begin{aligned}
2\mathbb{E} \left[ \langle \overline{\eta}^t \overline{y}_1^t, \overline{\eta}^t \overline{y}_1^{t-1} \rangle \right] &\leq 2a_7 \mathbb{E} \left[ \|\overline{\eta}^t \overline{y}_1^t\|^2 \right] + \frac{\eta_{\max}^2}{a_7 m} \sum_{i=1}^m \mathbb{E} \left[ \|\hat{y}_{1,i}^{t-1}\|^2 \right] \\
&+ a_8 \mathbb{E} \left[ \left( \frac{1}{m} \sum_{i=1}^m \frac{\eta_i^t}{\eta_i^{t-1}} \right)^2 \|\overline{x}^{t-1} - \overline{x}^t\|^2 \right] + \frac{2\beta^2}{a_7 m} \sum_{i=1}^m \mathbb{E} \left[ \|\hat{x}_i^{t-1}\|^2 \right] + \frac{2\sigma^2}{|\mathcal{B}|ma_8} \quad (24) \\
&+ \frac{1}{m} \sum_{i=1}^m \left( \frac{L^2 \eta_{\max}}{a_7} + \frac{2\beta^2}{a_7} + \frac{2L^2}{a_8} \right) \mathbb{E} \left[ \|\hat{x}_i^t\|^2 \right] + \frac{2}{m\beta} \sum_{i=1}^m \mathbb{E} \left[ \eta_i^t (f(\overline{x}^{t-1}) - f(\overline{x}^t)) \right].
\end{aligned}
$$

Substituting equation 11, equation 17, and equation 24 into equation 8, we obtain

$$
\begin{aligned}
(1 - 2a_7) \mathbb{E} \left[ \|\overline{\eta}^t \overline{y}_1^t\|^2 \right] &\leq (1 + a_2) \mathbb{E} \left[ \left\| \frac{1}{m} \sum_{i=1}^m \eta_i^t L_i^t (x_i^t - x_i^{t-1}) \right\|^2 \right] \\
&- ((1 - a_5)(1 - a_6)(1 - a_7) - a_8) \mathbb{E} \left[ \left( \frac{1}{m} \sum_{i=1}^m \frac{\eta_i^t}{\eta_i^{t-1}} \|\overline{x}^t - \overline{x}^{t-1}\| \right)^2 \right] \quad (25) \\
&+ \frac{2\beta}{m} \sum_{i=1}^m \mathbb{E} \left[ \eta_i^t (f(\overline{x}^{t-1}) - f(\overline{x}^t)) \right] + \delta_1^t,
\end{aligned}
$$

where the constant $\delta_1^t$ is given by

$$\delta_1^t = \frac{1}{m}\sum_{i=1}^m \left(\frac{L^2\eta_{\max}}{a_7} + \frac{2\beta^2}{a_7} + \frac{2L^2}{a_8}\right)\mathbb{E}\left[\|\hat{x}_i^t\|^2\right] + \frac{2\beta^2}{a_7 m}\sum_{i=1}^m \mathbb{E}\left[\|\hat{x}_i^{t-1}\|^2\right] + \frac{2\sigma^2}{|\mathcal{B}|ma_8}$$

$$+ \frac{\eta_{\max}^2}{a_7 m}\sum_{i=1}^m \mathbb{E}\left[\|\hat{y}_{1,i}^{t-1}\|^2\right] + 2\left(1-\frac{1}{a_4}\right)\frac{\eta_{\max}^2\sigma^2}{|\mathcal{B}|} + (1-a_5)\left(1-\frac{1}{a_6}\right)\frac{\eta_{\max}^2}{m}\sum_{i=1}^m \mathbb{E}\left[\|\hat{y}_{1,i}^{t-1}\|^2\right]$$

$$+ (1-a_4)(1-a_5)\left(1-\frac{1}{a_6}\right)\beta^2\frac{1}{m}\sum_{i=1}^m \mathbb{E}\left[\|\hat{x}_i^t(0)\|^2 + \rho^{2M}\|\hat{x}_i^{t-1}(0)\|^2\right]$$

$$+ \frac{2}{m}\sum_{i=1}^m \eta_{\max}^2\left(1+\frac{1}{a_2}\right)\mathbb{E}\left[\|\hat{y}_{1,i}^t\|^2 + \|\hat{y}_{2,i}^{t-1}\|^2\right]. \tag{26}$$

By Step 14 in Algorithm 1, we have

$$\eta_i^t \le \frac{7\sqrt{r}}{10}\frac{\eta_i^{t-1}}{\sqrt{[m(\eta_i^{t-1}L_i^t)^2 - 1]_+}}. \tag{27}$$

When $m(\eta_i^{t-1}L_i^t)^2 \le 1$,

$$(\eta_i^t L_i^t)^2 - \frac{(\eta_i^t)^2}{m(\eta_i^{t-1})^2} = (\eta_i^t)^2\left((L_i^t)^2 - \frac{1}{m(\eta_i^{t-1})^2}\right) \le 0 \le \frac{49r}{100}. \tag{28}$$

When $m(\eta_i^{t-1}L_i^t)^2 > 1$, equation 27 can be rewritten as

$$\eta_i^t \le \frac{7\sqrt{r}}{10}\frac{\eta_i^{t-1}}{\sqrt{m(\eta_i^{t-1}L_i^t)^2 - 1}}. \tag{29}$$

It implies that

$$(\eta_i^t L_i^t)^2 - \frac{(\eta_i^t)^2}{m(\eta_i^{t-1})^2} \le \frac{49r}{100m} \le \frac{49r}{100}. \tag{30}$$

According to equation 28 and equation 30, we always have

$$(\eta_i^t L_i^t)^2 - \frac{(\eta_i^t)^2}{m(\eta_i^{t-1})^2} \le \frac{49r}{100}. \tag{31}$$

Choose $a_2$, $a_5$, $a_6$, $a_7$, and $a_8$ such that the

$$a_2 \le \frac{1-r}{4\beta^2}, \ \max\{a_5, \ a_6, \ a_7, \ a_8\} \le \frac{47(1-r)}{1600\beta^2}. \tag{32}$$

Then we have

$$(1+a_2)(\eta_i^t L_i^t)^2 - ((1-a_5)(1-a_6)(1-a_7) - a_8)\frac{(\eta_i^t)^2}{m(\eta_i^{t-1})^2}$$

$$\le (1+a_2)\left((\eta_i^t L_i^t)^2 - \frac{((1-a_5)(1-a_6)(1-a_7) - a_8}{1+a_2}\frac{(\eta_i^t)^2}{m(\eta_i^{t-1})^2}\right)$$

$$\le (1+a_2)\left((\eta_i^t L_i^t)^2 - \frac{(\eta_i^t)^2}{m(\eta_i^{t-1})^2}\right) + ((1+a_2) - (1-a_5)(1-a_6)(1-a_7) + a_8)\beta^2$$

$$\le \frac{49}{100}\left(1 - \frac{3(1-r)}{4}\right) + \frac{147(1-r)}{400} \le \frac{49}{100}, \tag{33}$$

where we use the relationship $\frac{(\eta_i^t)^2}{(\eta_i^{t-1})^2} \le \beta^2$, $(1+a_2)\frac{49r}{100} = \frac{49}{100}r\left(1 + \frac{1-r}{4}\right) \le \frac{49}{100}\left(1 - \frac{3(1-r)}{4}\right)$, and

$$(1+a_2) - (1-a_5)(1-a_6)(1-a_7) + a_8 \le a_2 + a_5 + a_6 + a_7 + a_8$$

$$\le \frac{1-r}{4\beta^2} + \frac{47(1-r)}{400\beta^2} \le \frac{147(1-r)}{400\beta^2}.$$

Since $\frac{\eta_i^t}{\eta_i^{t-1}} \leq \beta$, it follows from equation 33 that

$$\eta_i^t L_i^t \leq \frac{1}{1+a_2}\left(((1-a_5)(1-a_6)(1-a_7) - a_8)\frac{(\eta_i^t)^2}{m(\eta_i^{t-1})^2} + \frac{49}{100}\right) < \frac{1}{1+a_2}\left(\beta^2 + \frac{1}{2}\right). \tag{34}$$

Hence, it follows from equation 27 that

$$(1+a_2)\mathbb{E}\left[\left\|\eta_i^t L_i^t(\bar{x}^t - \bar{x}^{t-1})\right\|^2\right] - \frac{(1-a_5)(1-a_6)(1-a_7) - a_8}{m}\mathbb{E}\left[\left(\frac{\eta_i^t}{\eta_i^{t-1}}\|\bar{x}^t - \bar{x}^{t-1}\|\right)^2\right]$$

$$\leq \frac{49}{100}\mathbb{E}[\|\bar{x}^t - \bar{x}^{t-1}\|^2]. \tag{35}$$

Applying the fact if $\|a_i\|^2 \leq \frac{1}{m}\|b_i\|^2 + \|c\|^2$ for all $i = 1, \ldots, m$, then $\|\bar{a}\|^2 \leq \|\bar{b}\|^2 + \|c\|^2$, where $a_i$, $b_i$, and $c$ are positive constants, $\bar{a} = \frac{1}{m}\sum_{i=1}^m a_i$, $\bar{b} = \frac{1}{m}\sum_{i=1}^m b_i$, , we have

$$(1+a_2)\mathbb{E}\left[\left\|\frac{1}{m}\sum_{i=1}^m \eta_i^t L_i^t(x_i^t - x_i^{t-1})\right\|^2\right]$$

$$- ((1-a_5)(1-a_6)(1-a_7) - a_8)\mathbb{E}\left[\left(\frac{1}{m}\sum_{i=1}^m \frac{\eta_i^t}{\eta_i^{t-1}}\|\bar{x}^t - \bar{x}^{t-1}\|\right)^2\right] \tag{36}$$

$$\leq \frac{24}{50}\mathbb{E}\left[\left\|\frac{1}{m}\sum_{i=1}^m(\bar{x}^t - \bar{x}^{t-1})\right\|^2\right] + 48(1+a_2)\frac{1}{m}\sum_{i=1}^m\mathbb{E}\left[\left\|\eta_i^t L_i^t(\hat{x}_i^t - \hat{x}_i^{t-1})\right\|^2\right].$$

By equation 34, we have

$$(1+a_2)\mathbb{E}\left[\left\|\eta_i^t L_i^t(\hat{x}_i^t - \hat{x}_i^{t-1})\right\|^2\right] \leq (2\beta^2 + 1)\mathbb{E}\left[\left\|\hat{x}_i^t\right\|^2 + \left\|\hat{x}_i^{t-1}\right\|^2\right]. \tag{37}$$

Based on the above analysis, we can rewrite equation 36 as follows:

$$(1-2a_8)\mathbb{E}\left[\|\overline{\eta}^t \overline{y}_1^t\|^2\right] \leq \frac{24}{50}\mathbb{E}\left[\left\|\bar{x}^t - \bar{x}^{t-1}\right\|^2\right] + \frac{2\beta^2+1}{m}\sum_{i=1}^m\mathbb{E}\left[\left\|\hat{x}_i^t\right\|^2 + \left\|\hat{x}_i^{t-1}\right\|^2\right]$$

$$+ \frac{2\beta}{m}\sum_{i=1}^m\mathbb{E}\left[\eta_i^t(f(\overline{x}^{t-1}) - f(\overline{x}^t))\right] + \delta_1^t. \tag{38}$$

By substituting equation 38 into equation 7, we obtain

$$\mathbb{E}\left[\|\overline{x}^{t+1} - \overline{x}^t\|^2\right]$$

$$\leq c_1\mathbb{E}\left[\|\overline{x}^t - \overline{x}^{t-1}\|^2\right] + \frac{2(1+a_1)}{m\beta(1-2a_8)}\sum_{i=1}^m\mathbb{E}\left[\eta_i^t(f(\overline{x}^{t-1}) - f(\overline{x}^t))\right] + \delta_3^t, \tag{39}$$

with $\delta_2^t = \frac{1+a_1}{1-2a_8}\left(\delta_1^t + \frac{2\beta^2+1}{m}\sum_{i=1}^m\mathbb{E}\left[\|\hat{x}_i^t\|^2 + \|\hat{x}_i^{t-1}\|^2\right]\right) + (1+\frac{1}{a_1})\frac{1}{m}\sum_{i=1}^m\eta_{\max}^2\|\hat{y}_{1,i}^t\|^2$ and $c_1 = \frac{1+a_1}{1-2a_8}\left(\frac{24(1+a_7)}{50}\right)$. Choose $a_8$ as

$$a_8 = \frac{1 - 124a_1}{250} > 0, \tag{40}$$

where $a_8$ exists due to $a_1 < \frac{1}{124}$ given in the lemma statement. Then we have

$$c_1 = \frac{24}{50} \times \frac{1+a_1}{1-2a_8} < \frac{1}{2}. \tag{41}$$

Substituting the second equation in equation 40 and equation 41 into equation 39, we obtain

$$
\mathbb{E}\left[\|\overline{x}^{t+1} - \overline{x}^t\|^2\right]
$$

$$
\leq c_1 \mathbb{E}\left[\|\overline{x}^t - \overline{x}^{t-1}\|^2\right] + \frac{125}{62m\beta} \sum_{i=1}^{m} \mathbb{E}\left[\eta_i^t (f(\overline{x}^{t-1}) - f(\overline{x}^t))\right] + \delta_3^t. \tag{42}
$$

Multiplying both sides of equation 42 by 2 leads to

$$
\mathbb{E}\left[\|\overline{x}^{t+1} - \overline{x}^t\|^2\right] \leq 2c_1 \mathbb{E}\left[\|\overline{x}^t - \overline{x}^{t-1}\|^2\right] - \mathbb{E}\left[\|\overline{x}^{t+1} - \overline{x}^t\|^2\right]
$$

$$
+ \frac{125}{31\beta}\mathbb{E}\left[\overline{\eta}^t (f(\overline{x}^{t-1}) - f(\overline{x}^t))\right] + 2\delta_3^t, \tag{43}
$$

which implies Lemma 1. $\qquad\square$

**Lemma 2.** *Under Assumptions 1 and 2, the following inequality holds for Algorithm 1:*

$$
\mathbb{E}[\|\overline{x}^{t+1} - x^*\|^2\} + \mathbb{E}[\|\overline{x}^{t+1} - \overline{x}^t\|^2\} + \left(2 + \frac{125\beta}{31}\right)\overline{\eta}^t \mathbb{E}[f(\overline{x}^t) - f(x^*)\}
$$

$$
\leq \left(1 - \frac{\mu}{2L} + a_9 \mu \eta_{\max}\right)\mathbb{E}[\|\overline{x}^t - x^*\|^2\} + \frac{45}{46}\mathbb{E}\left[\|\overline{x}^t - \overline{x}^{t-1}\|^2\right]
$$

$$
+ \gamma\left(2 + \frac{125\beta}{31}\right)\mathbb{E}\left[\overline{\eta}^{t-1}(f(\overline{x}^{t-1}) - f(x^*))\right] + \delta_5^t, \tag{44}
$$

*for any $\gamma \in (0,1)$, where the constant $\delta_5^t$ is given by*

$$
\delta_5^t = \frac{2\overline{b}_{\hat{x},1}}{m} \sum_{i=1}^{m} \mathbb{E}[\|\hat{x}_i^t\|^2] + \frac{\overline{b}_{\hat{x},2}}{m} \sum_{i=1}^{m} \mathbb{E}[\|\hat{x}_i^{t-1}\|^2] + \frac{\overline{b}_{\hat{x},2}}{m} \sum_{i=1}^{m} \mathbb{E}[\|\hat{x}_i^t(0)\|^2] + \frac{\overline{b}_{\hat{y},1}}{m} \sum_{i=1}^{m} \mathbb{E}[\|\hat{y}_{1,i}^t\|^2]
$$

$$
+ \frac{\overline{b}_{\hat{y},2}}{m} \sum_{i=1}^{m} \mathbb{E}[\|\hat{y}_{1,i}^{t-1}\|^2] + \frac{\overline{b}_{\hat{y},3}}{m} \sum_{i=1}^{m} \mathbb{E}[\|\hat{y}_{2,i}^{t-1}\|^2] + \frac{\overline{b}_\sigma \sigma^2}{|\mathcal{B}|}, \tag{45}
$$

*where $\overline{b}_{\hat{x},1} = 2b_{\hat{x},1} + \frac{2\eta_{\max}L^2}{a_9\mu}$, $\overline{b}_{\hat{x},2} = 2b_{\hat{x},2}$, $\overline{b}_{\hat{y},1} = 2b_{\hat{y},1} + + \frac{2\eta_{\max}L^2}{|a_9\mu}$, $\overline{b}_{\hat{y},2} = 2b_{\hat{y},2}$, $\overline{b}_{\hat{y},3} = 2b_{\hat{y},3}$, and $\overline{b}_\sigma = 2b_\sigma + \frac{2\eta_{\max}L^2}{|a_9\mu}$.*

*Proof.* According to the dynamics of $x_i^t$ in Algorithm 1, we have

$$
\mathbb{E}\left[\|\overline{x}^{t+1} - x^*\|^2\right] = \mathbb{E}\left[\|\overline{x}^t - \overline{\eta^t y^t} - x^*\|^2\right]
$$

$$
= \mathbb{E}\left[\|\overline{x}^t - x^*\|^2\right] + \mathbb{E}\left[\|\overline{\eta^t y^t}\|^2\right] - 2\mathbb{E}\left[\langle\overline{x}^t - x^*, \overline{\eta^t y^t}\rangle\right] \tag{46}
$$

$$
= \mathbb{E}\left[\|\overline{x}^t - x^*\|^2\right] + \mathbb{E}\left[\|\overline{x}^{t+1} - \overline{x}^t\|^2\right] - 2\mathbb{E}\left[\langle\overline{x}^t - x^*, \overline{\eta^t y^t}\rangle\right],
$$

with $\overline{\eta^t y^t} := \frac{1}{N}\sum_{i=1}^{N}\eta_i^t y_{1,i}^t$.

The third term on the right-hand side of equation 46 satisfies:

$$
2\mathbb{E}\left[\langle\overline{x}^t - x^*, \overline{\eta^t y^t}\rangle\right] = -2\langle\overline{x}^t - x^*, \frac{1}{m}\sum_{i=1}^{m}\eta_i^t y_{1,i}^t\rangle
$$

$$
\leq -2\mathbb{E}\left[\overline{\eta}^t(f(\overline{x}^t) - f(x^*)) - \mu\eta_{\min}\|\overline{x}^t - x^*\|^2\right] \tag{47}
$$

$$
- 2\mathbb{E}\left[\left\langle\overline{x}^t - x^*, \frac{1}{m}\sum_{i=1}^{m}\eta_i^t(y_{1,i}^t - \nabla f(\overline{x}^t))\right\rangle\right],
$$

where in the derivation we have used the $\mu$-strong convexity of $f(x)$ and $\eta_{\min} = \min_{i\in[m],t\in\mathbb{N}^+}\eta_i^t$. Furthermore, since the function $f$ is L-smoothness, the minimum of the stepsizes exists.

By using the Cauchy–Schwarz inequality, the third term on the right-hand side of inequality equation 47 satisfies

$$- 2\mathbb{E}\left[\left\langle \overline{x}^t - x^*, \frac{1}{m}\sum_{i=1}^m \eta_i^t(y_{1,i}^t - \nabla f(\overline{x}^t))\right\rangle\right]$$

$$= -\frac{2}{m}\sum_{i=1}^m \mathbb{E}\left[\left\langle \sqrt{\eta_i^t}(\overline{x}^t - x^*), \sqrt{\eta_i^t}(y_{1,i}^t - \nabla f(\overline{x}^t))\right\rangle\right]$$

$$\leq \frac{2}{m}\sqrt{\left(\sum_{i=1}^m \mathbb{E}\left[\left\|\sqrt{\eta_i^t}(\overline{x}^t - x^*)\right\|^2\right]\right)\left(\sum_{i=1}^m \mathbb{E}\left[\left\|\sqrt{\eta_i^t}(y_{1,i}^t - \nabla f(\overline{x}^t))\right\|^2\right]\right)}.$$

By applying the inequality $2\langle a, b\rangle \leq \alpha\|a\|^2 + \frac{1}{\alpha}\|b\|^2$ for any $\alpha > 0$ and $a$, $b \in \mathbb{R}^n$ to equation 48, we obtain

$$- 2\mathbb{E}\left[\left\langle \overline{x}^t - x^*, \frac{1}{m}\sum_{i=1}^m \eta_i^t(y_{1,i}^t - \nabla f(\overline{x}^t))\right\rangle\right]$$

$$\leq \frac{a_9\mu}{m}\sum_{i=1}^m \mathbb{E}\left[\eta_i^t\|\overline{x}^t - x^*\|^2\right] + \frac{1}{ma_9\mu}\sum_{i=1}^m \mathbb{E}\left[\eta_i^t\|y_{1,i}^t - \nabla f(\overline{x}^t)\|^2\right]$$

$$\leq a_9\mu\eta_{\max}\mathbb{E}\left[\|\overline{x}^t - x^*\|^2\right] + \frac{\eta_{\max}}{ma_9\mu}\sum_{i=1}^m \mathbb{E}\left[\|y_{1,i}^t - \nabla f(\overline{x}^t)\|^2\right]$$

for any positive $a_9$.

The second term on the right-hand side of equation 48 satisfies

$$\frac{\eta_{\max}}{ma_9\mu}\sum_{i=1}^m \mathbb{E}\left[\|y_{1,i}^t - \nabla f(\overline{x}^t)\|^2\right] = \frac{\eta_{\max}}{ma_9\mu}\sum_{i=1}^m \mathbb{E}\left[\|y_{1,i}^t - \overline{y}_1^t + \overline{y}_1^t - \nabla f(\overline{x}^t)\|^2\right]$$

$$\leq \frac{2\eta_{\max}}{ma_9\mu}\sum_{i=1}^m \mathbb{E}\left[\|y_{1,i}^t - \overline{y}_1^t\|^2\right] + \frac{2\eta_{\max}}{ma_9\mu}\sum_{i=1}^m \mathbb{E}\left[\|\overline{y}_1^t - \nabla f(\overline{x}^t)\|^2\right]. \tag{48}$$

By using the Lipschitz continuity of $\nabla f$ from Assulption 1, we have

$$\sum_{i=1}^m \mathbb{E}\left[\|\overline{y}_1^t - \nabla f(\overline{x}^t)\|^2\right] \leq \frac{1}{m}\sum_{i=1}^m \mathbb{E}\left[\|g_i^t(x_i^t) - \nabla f_i(\overline{x}^t)\|^2\right]$$

$$= \frac{\sigma^2}{B} + \frac{1}{m}\sum_{i=1}^m \mathbb{E}\left[\|\nabla f_i^t(x_i^t) - \nabla f_i(\overline{x}^t)\|^2\right] \leq \frac{\sigma^2}{|\mathcal{B}|} + \frac{L^2}{m}\sum_{i=1}^m \mathbb{E}\left[\|\hat{x}_i^t\|^2\right].$$

Substituting equation 49 into equation 48 leads to

$$\frac{\eta_{\max}}{ma_9\mu}\sum_{i=1}^m \mathbb{E}\left[\|y_{1,i}^t - \nabla f(\overline{x}^t)\|^2\right]$$

$$\leq \frac{2\eta_{\max}}{ma_9\mu}\sum_{i=1}^m \mathbb{E}\left[\|\hat{y}_{1,i}^t\|^2\right] + \frac{2\eta_{\max}L^2}{ma_9\mu}\sum_{i=1}^m \mathbb{E}\left[\|\hat{x}_i^t\|^2\right] + \frac{2\eta_{\max}L^2\sigma^2}{|\mathcal{B}|a_9\mu}. \tag{49}$$

By substituting equation 47 to equation 49 into equation 46, we obtain

$$\mathbb{E}\left[\|\overline{x}^{t+1} - x^*\|^2\right] \leq (1 - \mu\eta_{\min} + a_9\mu\eta_{\max})\mathbb{E}\left[\|\overline{x}^t - x^*\|^2\right]$$

$$+ \mathbb{E}\left[\|\overline{x}^{t+1} - \overline{x}^t\|^2\right] - 2\mathbb{E}\left[\overline{\eta}^t(f(\overline{x}^t) - f(x^*))\right] + \delta_4^t, \tag{50}$$

with $\delta_4^t = \frac{2\eta_{\max}}{ma_9\mu}\sum_{i=1}^m \mathbb{E}\left[\|\hat{y}_{1,i}^t\|^2\right] + \frac{2\eta_{\max}L^2}{ma_9\mu}\sum_{i=1}^m \mathbb{E}\left[\|\hat{x}_i^t\|^2\right] + \frac{2\eta_{\max}L^2\sigma^2}{|\mathcal{B}|a_9\mu}$.

By adding both sides of equation 4 in Lemma 1 and equation 50, we obtain

$$
\mathbb{E}\left[\|\overline{x}^{t+1} - x^*\|^2\right] + \mathbb{E}\left[\|\overline{x}^{t+1} - \overline{x}^t\|^2\right] + \left(2 + \frac{125\beta}{31}\right)\mathbb{E}\left[\overline{\eta}^t(f(\overline{x}^t) - f(x^*))\right]
$$

$$
\leq (1 - \mu\eta_{\min} + a_9\mu\eta_{\max})\mathbb{E}\left[\|\overline{x}^t - x^*\|^2\right] + \frac{45}{46}\mathbb{E}\left[\|\overline{x}^t - \overline{x}^{t-1}\|^2\right] \tag{51}
$$

$$
+ \frac{125\beta}{31}\mathbb{E}\left[\overline{\eta}^t(f(\overline{x}^{t-1}) - f(x^*))\right] + \delta_5^t,
$$

with $\delta_5^t = \delta_4^t + 2\delta_3^t$.

By setting $\beta \in (1, 1.36)$ and using Line 16 in Algorithm 1, we have $\frac{125\beta}{31}\overline{\eta}^t \leq \frac{125\beta^2}{31}\overline{\eta}^{t-1} = \gamma_1\left(2 + \frac{125\beta}{31}\right)\overline{\eta}^{t-1}$ for some $\gamma_1 \in \left(0, \frac{91}{92}\right)$, which implies the following inequality:

$$
\frac{125\beta}{31}\overline{\eta}^t \leq \gamma_1\left(2 + \frac{125\beta}{31}\right)\overline{\eta}^{t-1} \quad \text{and} \quad \frac{125\beta^2}{31} \leq \gamma_1\left(2 + \frac{125\beta}{31}\right). \tag{52}
$$

By letting $a_9 = \frac{\eta_{\min}}{2\eta_{\max}}$, we have

$$
\mathbb{E}\left[\|\overline{x}^{t+1} - x^*\|^2\right] + \mathbb{E}\left[\|\overline{x}^{t+1} - \overline{x}^t\|^2\right] + \left(2 + \frac{125\beta}{31}\right)\mathbb{E}\left[\overline{\eta}^t(f(\overline{x}^t) - f(x^*))\right]
$$

$$
\leq \left(1 - \frac{\mu\eta_{\min}}{2}\right)\mathbb{E}\left[\|\overline{x}^t - x^*\|^2\right] + \frac{45}{46}\mathbb{E}\left[\|\overline{x}^t - \overline{x}^{t-1}\|^2\right] \tag{53}
$$

$$
+ \mathbb{E}\left[\gamma_1\left(2 + \frac{125\beta}{31}\right)\overline{\eta}^{t-1}(f(\overline{x}^{t-1}) - f(x^*))\right] + \delta_5^t,
$$

which proves Lemma 2. $\qquad\square$

**Lemma 3.** *Under Assumptions 1 and 2, the following inequality holds for Algorithm 1:*

$$
\sum_{i=1}^m \mathbb{E}\left[\|\hat{x}_i^t\|^2\right] \leq \rho^{2M}\left(12\eta_{\max}^2\sum_{i=1}^m\mathbb{E}\left[\|\hat{y}_{1,i}^{t-1}\|^2\right] + (24\eta_{\max}^2 L^2 + 3)\sum_{i=1}^m\mathbb{E}\left[\|\hat{x}_i^{t-1}\|^2\right]\right.
$$

$$
\left. + 48m\eta_{\max}^2 L^2\mathbb{E}\left[\|\overline{x}^t - x^*\|^2\right] + 48m\eta_{\max}^2 L^2\mathbb{E}\left[\|\overline{x}^t - \overline{x}^{t-1}\|^2\right] + \frac{12\eta_{\max}^2\sigma^2}{|\mathcal{B}|}\right), \tag{54}
$$

$$
\sum_{i=1}^m \mathbb{E}\left[\|\hat{y}_{1,i}^t\|^2\right] \leq \rho^{2M}\left(18L^2\sum_{i=1}^m\mathbb{E}\left[\|\hat{x}_i^t\|^2\right] + 18L^2\sum_{i=1}^m\mathbb{E}\left[\|\hat{x}_i^{t-1}\|^2\right] + 3\sum_{i=1}^m\mathbb{E}\left[\|\hat{y}_{1,i}^{t-1}\|^2\right]\right.
$$

$$
\left. + 18mL^2\mathbb{E}\left[\|\overline{x}^t - \overline{x}^{t-1}\|^2\right] + \frac{36m\sigma^2}{|\mathcal{B}|}\right), \tag{55}
$$

$$
\sum_{i=1}^m \mathbb{E}\left[\|\hat{y}_{2,i}^t\|^2\right] \leq \rho^{2M}\left(18L^2\sum_{i=1}^m\mathbb{E}\left[\|\hat{x}_i^t\|^2\right] + 18L^2\sum_{i=1}^m\mathbb{E}\left[\|\hat{x}_i^{t-1}\|^2\right] + 3\sum_{i=1}^m\mathbb{E}\left[\|\hat{y}_{2,i}^{t-1}\|^2\right]\right.
$$

$$
\left. + 18mL^2\mathbb{E}\left[\|\overline{x}^t - \overline{x}^{t-1}\|^2\right] + \frac{36m\sigma^2}{|\mathcal{B}|}\right), \tag{56}
$$

*where $\rho < 1$ is from Assumption 2 and $M$ is the number of inner-consensus-loop iterations from Algorithm 1.*

*Proof.* According to Line 5 in Algorithm 1, we have

$$
\mathbf{X}^t(q+1) = (W \otimes I_n)\mathbf{X}^t(q), \ q = 0, 1, \ldots, M-1, \tag{57}
$$

where $W \in \mathbb{R}^{m\times m}$ is the adjacency matrix given in Assumption 2. Since the relationship $\overline{x}^t(q) = \frac{1}{m}\sum_{i=1}^m x_i^t(q)$ holds, we have

$$
\overline{x}^t(q+1) = \frac{1}{m}\sum_{i=1}^m x_i^t(q+1) = \frac{1}{m}\sum_{i=1}^m\sum_{j=1}^m w_{ij}x_j^t(q) = \frac{1}{m}\sum_{j=1}^m x_j^t(q) = \overline{x}^t(q), \tag{58}
$$

where we have used Assumption 2 in the derivation.

By using the definition $\bar{\mathbf{X}}^t = \text{col}(\overline{x}^t, \cdots, \overline{x}^t) \in \mathbb{R}^{mn}$ and equation 58, we have

$$\bar{\mathbf{X}}^t = (W \otimes I_n)\bar{\mathbf{X}}^t. \tag{59}$$

By defining $\Delta^t(q) \triangleq \mathbf{X}^t(q) - \bar{\mathbf{X}}^t$ and subtracting equation 59 from equation 57, we obtain

$$\Delta^t(q+1) = \mathbf{X}^t(q+1) - \mathbf{X}_1^t = (W \otimes I_n)\Delta^t(q)$$
$$\Delta^t(q+1) = (W \otimes I_n)\Delta^t(q).$$

Since $W$ is a doubly stochastic matrix, there must exist an orthogonal matrix $\Phi \in \mathbb{R}^{m \times m}$ such that $W$ satisfies the following transformation:

$$\Phi^\top W \Phi = \text{diag}\{1, \lambda_2, \ldots, \lambda_m\}, \tag{60}$$

with $|\lambda_i| < 1$, $i = 2, \ldots, m$. The first column of $\Phi$ is given by $\frac{1}{\sqrt{m}}\mathbf{1}_n$, which corresponds to the eigenvalue 1 of $W$. By further considering the following transformation:

$$\Delta_1^t(q) = (\Phi^\top \otimes I_n)\Delta^t(q), \tag{61}$$

with $\Delta_1^t(q) = [\sigma_1^t(q); \sigma_2^t(q); \ldots; \sigma_m^t(q)] \in \mathbb{R}^{mn}$, we have

$$\sigma_i^t(q) = \sum_{j=1}^{m} \Phi_{ij}^\top (x_j^t(q) - \overline{x}^t), \tag{62}$$

where $\Phi_{ij}^\top$ denotes the element in the $i$th row and $j$th column of the matrix $\Phi^\top$. By using $\sigma_1^t(q) = \frac{1}{\sqrt{m}}\sum_{j=1}^{m}(x_j^t(q) - \overline{x}^t) = \mathbf{0}$, equation 60 can be rewritten as follows:

$$\Delta_1^t(q+1) = (\text{diag}\{1, \lambda_2, \ldots, \lambda_m\} \otimes I_n)\Delta_1^t(q). \tag{63}$$

Since the relationship $\sigma_1^t(q) = 0$ holds, equation 63 implies

$$\sigma_i^t(q+1) = \lambda_i \sigma_i^t(q) \le \rho \sigma_i^t(q) \le \rho^{q+1}\sigma_i^t(0), \tag{64}$$

with $\rho = \max\{|\lambda_2|, \cdots, |\lambda_m|\} < 1$. According to equation 64, we have

$$\|\Delta^t(M)\|^2 \le \rho^{2M}\|\Delta^t(0)\|^2, \tag{65}$$

which further implies

$$\sum_{i=1}^{m} \|x_i^t - \overline{x}^t\|^2 \le \rho^{2M} \sum_{i=1}^{m} \|x_i^t(0) - \overline{x}^t\|^2. \tag{66}$$

By using an argument similar to the derivation of equation 66, we obtain

$$\sum_{i=1}^{m} \|y_{1,i}^t - \overline{y}_1^t\|^2 \le \rho^{2M} \sum_{i=1}^{m} \|y_{1,i}^t(0) - \overline{y}_1^t\|^2,$$
$$\sum_{i=1}^{m} \|y_{2,i}^t - \overline{y}_2^t\|^2 \le \rho^{2M} \sum_{i=1}^{m} \|y_{2,i}^t(0) - \overline{y}_2^t\|^2. \tag{67}$$

Using equation 66, we have

$$\begin{aligned}
\sum_{i=1}^{m} \|x_i^t - \overline{x}^t\|^2 &\le \rho^{2M} \sum_{i=1}^{m} \|x_i^t(0) - \overline{x}^t\|^2 \\
&= \rho^{2M} \sum_{i=1}^{m} \|x_i^t(0) - x_i^{t-1} + x_i^{t-1} - \overline{x}^{t-1} + \overline{x}^{t-1} - \overline{x}^t\|^2 \\
&\le 3\rho^{2M}\left(\sum_{i=1}^{m} \|x_i^t(0) - x_i^{t-1}\|^2 + \sum_{i=1}^{m} \|x_i^{t-1} - \overline{x}^{t-1}\|^2 + \sum_{i=1}^{m} \|\overline{x}^{t-1} - \overline{x}^t\|^2\right) \\
&= 3\rho^{2M}\left(\sum_{i=1}^{m} \|x_i^t(0) - x_i^{t-1}\|^2 + \sum_{i=1}^{m} \|\hat{x}_i^{t-1}\|^2 + m\|\overline{x}^t - \overline{x}^{t-1}\|^2\right),
\end{aligned} \tag{68}$$

where we have used the relationship $\|a+b+c\|^2 \leq 3\|a\|^2 + 3\|b\|^2 + 3\|c\|^2$ in the second inequality.

We estimate an upper bound on the first term on the right-hand side of equation 68 as follows:

$$\sum_{i=1}^{m} \|x_i^t(0) - x_i^{t-1}\|^2 = \sum_{i=1}^{m} \|\eta_i^{t-1} y_{1,i}^{t-1}\|^2 \leq 2\sum_{i=1}^{m} \|\eta_i^{t-1} \hat{y}_{1,i}^{t-1}\|^2 + 2\sum_{i=1}^{m} \|\eta_i^{t-1} \overline{y}_1^{t-1}\|^2$$

$$= 2\eta_{\max}^2 \sum_{i=1}^{m} \|\hat{y}_{1,i}^{t-1}\|^2 + 2m\eta_{\max}^2 \left\| \frac{1}{m} \sum_{i=1}^{m} (g_i^{t-1}(x_i^{t-1}) - \nabla f(x^*)) \right\|^2. \tag{69}$$

By using the following inequality and equation 69

$$\mathbb{E}\left[ \left\| \frac{1}{m} \sum_{i=1}^{m} (g_i^{t-1}(x_i^{t-1}) - \nabla f(x^*)) \right\|^2 \right] \leq \frac{\sigma^2}{|\mathcal{B}|m} + \frac{1}{m} \mathbb{E}\left[ \left\| \sum_{i=1}^{m} (\nabla f_i(x_i^{t-1}) - \nabla f_i(x^*)) \right\|^2 \right]$$

$$\leq \frac{\sigma^2}{|\mathcal{B}|m} + \frac{L^2}{m} \sum_{i=1}^{m} \mathbb{E}\left[ \|x_i^{t-1} - x^*\|^2 \right],$$

we obtain the following relationship:

$$\sum_{i=1}^{m} \mathbb{E}\left[ \|x_i^t(0) - x_i^{t-1}\|^2 \right]$$

$$\leq 2\eta_{\max}^2 \sum_{i=1}^{m} \mathbb{E}\left[ \|\hat{y}_{1,i}^{t-1}\|^2 \right] + 2\eta_{\max}^2 L^2 \sum_{i=1}^{m} \mathbb{E}\left[ \|x_i^{t-1} - x^*\|^2 \right] + 2\eta_{\max}^2 \sigma^2$$

$$\leq 2\eta_{\max}^2 \sum_{i=1}^{m} \|\hat{y}_{1,i}^{t-1}\|^2 + 4\eta_{\max}^2 L^2 \sum_{i=1}^{m} \mathbb{E}\left[ \|\hat{x}_i^{t-1}\|^2 \right] \tag{70}$$

$$+ 4m\eta_{\max}^2 L^2 \mathbb{E}\left[ \|\overline{x}^{t-1} - x^*\|^2 \right] + 2\eta_{\max}^2 \sigma^2$$

$$\leq 2\eta_{\max}^2 \sum_{i=1}^{m} \mathbb{E}\left[ \|\hat{y}_{1,i}^{t-1}\|^2 \right] + 4\eta_{\max}^2 L^2 \sum_{i=1}^{m} \mathbb{E}\left[ \|\hat{x}_i^{t-1}\|^2 \right] + 8m\eta_{\max}^2 L^2 (\mathbb{E}\left[ \|\overline{x}^t - \overline{x}^{t-1}\|^2 \right]$$

$$+ \mathbb{E}\left[ \|\overline{x}^t - x^*\|^2 \right]) + \frac{2\eta_{\max}^2 \sigma^2}{|\mathcal{B}|}.$$

The third term on the right-hand side of equation 68 satisfies

$$m\|\overline{x}^{t-1} - \overline{x}^t\|^2$$

$$= m\|\overline{\eta^{t-1} y^{t-1}}\|^2 = m\| \frac{1}{m} \sum_{i=1}^{m} \eta_i^{t-1} y_i^{t-1}\|^2$$

$$\leq \sum_{i=1}^{m} \|\eta_i^{t-1} y_i^{t-1}\|^2 \leq \eta_{\max}^2 \sum_{i=1}^{m} \|y_i^{t-1}\|^2 \leq 2\eta_{\max}^2 \sum_{i=1}^{m} \|\hat{y}_{1,i}^{t-1}\|^2 + 2m\eta_{\max}^2 \|\overline{y}_1^{t-1}\|^2 \tag{71}$$

$$= 2\eta_{\max}^2 \left( \sum_{i=1}^{m} \|\hat{y}_{1,i}^{t-1}\|^2 + m \left\| \frac{1}{m} \sum_{i=1}^{m} (g_i^{t-1}(x_i^{t-1}) - f(x^*)) \right\|^2 \right),$$

with $\overline{\eta^{t-1} y^{t-1}} = \frac{1}{m} \sum_{i=1}^{m} \eta_i^{t-1} y_{1,i}^{t-1}$. Substituting equation 69 into equation 72 leads to

$$m\mathbb{E}\left[ \|\overline{x}^{t-1} - \overline{x}^t\|^2 \right]$$

$$\leq 2\eta_{\max}^2 \left( \sum_{i=1}^{m} \mathbb{E}\left[ \|\hat{y}_{1,i}^{t-1}\|^2 \right] + L^2 \sum_{i=1}^{m} \mathbb{E}\left[ \|x_i^{t-1} - x^*\|^2 \right] \right) + 2\eta_{\max}^2 \sigma^2$$

$$\leq 2\eta_{\max}^2 \sum_{i=1}^{m} \mathbb{E}\left[ \|\hat{y}_{1,i}^{t-1}\|^2 \right] + 4\eta_{\max}^2 L^2 \sum_{i=1}^{m} \mathbb{E}\left[ \|\hat{x}_i^{t-1}\|^2 \right] \tag{72}$$

$$+ 8m\eta_{\max}^2 L^2 \left( \mathbb{E}\left[ \|\overline{x}^t - \overline{x}^{t-1}\|^2 \right] + \mathbb{E}\left[ \|\overline{x}^t - x^*\|^2 \right] \right) + \frac{2\eta_{\max}^2 \sigma^2}{|\mathcal{B}|}.$$

By substituting equation 70 and equation 72 into equation 68, we arrive at equation 54.

By using equation 67, we have

$$
\begin{aligned}
&\sum_{i=1}^{m} \|y_{1,i}^t - \overline{y}_1^t\|^2 \\
&\leq \rho^{2M} \sum_{i=1}^{m} \|y_{1,i}^t(0) - \overline{y}_1^t\|^2 \\
&= \rho^{2M} \sum_{i=1}^{m} \|y_{1,i}^t(0) - y_{1,i}^{t-1} + y_{1,i}^{t-1} - \overline{y}_1^{t-1} + \overline{y}_1^{t-1} - \overline{y}_1^t\|^2 \\
&\leq 3\rho^{2M} \left( \sum_{i=1}^{m} \|y_{1,i}^t(0) - y_{1,i}^{t-1}\|^2 + \sum_{i=1}^{m} \|\hat{y}_{1,i}^{t-1}\|^2 + m\|\overline{y}_1^t - \overline{y}_1^{t-1}\|^2 \right).
\end{aligned}
\tag{73}
$$

The first term on the right-hand side of equation 73 satisfies

$$
\begin{aligned}
\sum_{i=1}^{m} \mathbb{E}\left[\|y_{1,i}^t(0) - y_{1,i}^{t-1}\|^2\right] &= \sum_{i=1}^{m} \mathbb{E}\left[\|g_i^{t-1}(x_i^t) - g_i^{t-2}(x_i^{t-1})\|^2\right] \\
&\leq \sum_{i=1}^{m} \mathbb{E}\left[\|\nabla f_i(x_i^t) - \nabla f_i(x_i^{t-1})\|^2\right] + \frac{2m\sigma^2}{|\mathcal{B}|} \\
&\leq L^2 \sum_{i=1}^{m} \mathbb{E}\left[\|x_i^t - x_i^{t-1}\|^2\right] + \frac{2m\sigma^2}{|\mathcal{B}|} \\
&= L^2 \sum_{i=1}^{m} \mathbb{E}\left[\|x_i^t - \overline{x}^t + \overline{x}^t - \overline{x}^{t-1} + \overline{x}^{t-1} - x_i^{t-1}\|^2\right] + \frac{2m\sigma^2}{|\mathcal{B}|} \\
&\leq 3L^2 \left( \sum_{i=1}^{m} \mathbb{E}\left[\|\hat{x}_i^t\|^2\right] + m\mathbb{E}\left[\|\overline{x}^t - \overline{x}^{t-1}\|^2\right] + \sum_{i=1}^{m} \mathbb{E}\left[\|\hat{x}_i^{t-1}\|^2\right] \right) + \frac{2m\sigma^2}{|\mathcal{B}|}.
\end{aligned}
\tag{74}
$$

The third term on the right-hand side of inequality equation 73 satisfies

$$
\begin{aligned}
m\mathbb{E}\left[\|\overline{y}_1^t - \overline{y}_1^{t-1}\|^2\right] &= m\mathbb{E}\left[\left\|\frac{1}{m}\sum_{i=1}^{m}(g_i^{t-1}(x_i^t) - g_i^{t-2}(x_i^{t-1}))\right\|^2\right] \\
&= m\mathbb{E}\left[\left\|\frac{1}{m}\sum_{i=1}^{m}(\nabla f_i(x_i^t) - \nabla f_i(x_i^{t-1}))\right\|^2\right] + \frac{2\sigma^2}{|\mathcal{B}|} \leq L^2 \sum_{i=1}^{m} \mathbb{E}\left[\|x_i^t - x_i^{t-1}\|^2\right] + \frac{2\sigma^2}{|\mathcal{B}|} \\
&\leq 3L^2 \left( \sum_{i=1}^{m} \mathbb{E}\left[\|\hat{x}_i^t\|^2\right] + m\mathbb{E}\left[\|\overline{x}^t - \overline{x}^{t-1}\|^2\right] + \sum_{i=1}^{m} \mathbb{E}\left[\|\hat{x}_i^{t-1}\|^2)\right] \right) + \frac{2\sigma^2}{|\mathcal{B}|}.
\end{aligned}
$$

By substituting equation 74 and equation 75 into equation 73, we arrive at equation 55.

The proof of equation 56 is similar to the derivation of equation 55, and thus is omitted here. $\square$

### B.2 PROOF OF THEOREM 1

*Proof of theorem 1:* By setting $\alpha_1 = 1 - \frac{\mu}{2L}$, $\alpha_2 = \min\left\{\frac{45}{46}, \frac{1-r}{4\beta^2}\right\}$, $\alpha_3 = \frac{125}{62m}\left(\frac{L^2\eta_{\max}}{a_6} + \frac{2\beta^2}{a_8} + b_1 + \frac{49(1+a_7)}{50a_7} + \frac{124L^2\eta_{\max}}{125a_9\mu}\right)$, $\alpha_4 = \frac{125}{62m}\left(\frac{2\beta^2}{a_8} + b_1 + \frac{49(1+a_7)}{50a_7}\right)$, $\alpha_5 = \frac{\eta_{\max}^2}{m}\left(\frac{187}{31} + \frac{125}{31a_2} + \frac{2}{a_1} + \frac{2}{a_9\mu\eta_{\max}}\right)$, $\alpha_6 = \frac{125\eta_{\max}^2}{62m}\left(\frac{1}{a_3} + 1 + \frac{1}{a_8} + \frac{2}{a_2}\right)$, and $\alpha_7 := 2\left(\left(1 - \frac{1}{a_3}\right)\left(2 + \frac{2\eta_{\max}}{\eta_{\min}} + \frac{\eta_{\min}}{2\eta_{\max}}\right) + \frac{1}{a_6m}\right) + \frac{2\eta_{\max}L^2}{a_9\mu}$,

equation 44 can be rewritten as follows:

$$
\mathbb{E}[\|\overline{x}^{t+1} - x^*\|^2\} + \mathbb{E}[\|\overline{x}^{t+1} - \overline{x}^t\|^2\} + \left(2 + \frac{125\beta}{31}\right)\mathbb{E}[\overline{\eta}^t(f(\overline{x}^t) - f(x^*))\}
$$

$$
\leq \alpha_1\mathbb{E}[\|\overline{x}^t - x^*\|^2\} + \alpha_2\mathbb{E}[\|\overline{x}^t - \overline{x}^{t-1}\|^2\} + \gamma\left(2 + \frac{125\beta}{31}\right)\mathbb{E}[\overline{\eta}^{t-1}(f(\overline{x}^{t-1}) - f(x^*))]
$$

$$
+ \alpha_3\sum_{i=1}^m \mathbb{E}[\|\hat{x}_i^t\|^2] + \alpha_4\sum_{i=1}^m \mathbb{E}[\|\hat{x}_i^{t-1}\|^2] + \alpha_5\sum_{i=1}^m \left(\mathbb{E}[\|\hat{y}_{1,i}^t\|^2] + \mathbb{E}[\|\hat{y}_{2,i}^{t-1}\|^2]\right) \tag{75}
$$

$$
+ \alpha_6\sum_{i=1}^m \mathbb{E}[\|\hat{y}_{1,i}^{t-1}\|^2] + \frac{\alpha_7\sigma^2}{|\mathcal{B}|}.
$$

By using an argument similar to the derivation of equation 75, equation 54 and equation 55 can be rewritten as follows:

$$
\sum_{i=1}^m \mathbb{E}[\|\hat{x}_i^t\|^2\} \leq \rho^{2M}\left(\alpha_8\sum_{i=1}^m \mathbb{E}[\|\hat{y}_{1,i}^{t-1}\|^2] + \alpha_9\sum_{i=1}^m \mathbb{E}[\|\hat{x}_i^{t-1}\|^2]\right.
$$

$$
\left. + \alpha_{10}\mathbb{E}[\|\overline{x}^t - x^*\|^2] + \alpha_{10}\mathbb{E}[\|\overline{x}^t - \overline{x}^{t-1}\|^2] + \frac{\alpha_8\sigma^2}{|\mathcal{B}|}\right), \tag{76}
$$

$$
\sum_{i=1}^m \left(\mathbb{E}[\|\hat{y}_{1,i}^t\|^2\} + \mathbb{E}[\|\hat{y}_{2,i}^t\|^2]\right) \leq \rho^{2M}(\alpha_{11}\sum_{i=1}^m \mathbb{E}[\|\hat{x}_i^t\|^2] + \alpha_{11}\sum_{i=1}^m \mathbb{E}[\|\hat{x}_i^{t-1}\|^2]
$$

$$
+ \alpha_{12}\sum_{i=1}^m \left(\mathbb{E}[\|\hat{y}_{1,i}^{t-1}\|^2\} + \mathbb{E}[\|\hat{y}_{2,i}^{t-1}\|^2]\right) + \alpha_{13}\mathbb{E}[\|\overline{x}^t - \overline{x}^{t-1}\|^2] + \frac{\alpha_{14}\sigma^2}{|\mathcal{B}|}, \tag{77}
$$

with $\alpha_8 = 12\eta_{\max}^2$, $\alpha_9 = 24\eta_{\max}^2 L^2 + 3$, $\alpha_{10} = 48m\eta_{\max}^2 L^2$, $\alpha_{11} = 18L^2$, $\alpha_{12} = 3$, $\alpha_{13} = 18mL^2 > 0$, and $\alpha_{14} = 72m$.

Multiplying inequalities equation 76 and equation 77 by $K$ and then using equation 75 lead to

$$
\mathbb{E}[\|\overline{x}^{t+1} - x^*\|^2] + \mathbb{E}[\|\overline{x}^{t+1} - \overline{x}^t\|^2] + \left(2 + \frac{125\beta}{31}\right)\mathbb{E}[\overline{\eta}^t(f(\overline{x}^t) - f(x^*))]
$$

$$
+ (K - \alpha_3 - \rho^{2M}\alpha_{10}K)\sum_{i=1}^m \mathbb{E}[\|\hat{x}_i^t\|^2] + (K - \alpha_5)\sum_{i=1}^m \left(\mathbb{E}[\|\hat{y}_{1,i}^t\|^2] + \mathbb{E}[\|\hat{y}_{2,i}^t\|^2]\right)
$$

$$
\leq \left(\alpha_1 + \rho^{2M}\alpha_9 K\right)\mathbb{E}[\|\overline{x}^t - x^*\|^2] + (\alpha_2 + \rho^{2M}K(\alpha_9 + \alpha_{12}))\mathbb{E}[\|\overline{x}^t - \overline{x}^{t-1}\|^2]
$$

$$
+ \gamma(2 + \frac{125\beta}{31})\mathbb{E}[\overline{\eta}^{t-1}(f(\overline{x}^{t-1}) - f(x^*))] + \left(\alpha_4 + \rho^{2M}K(\alpha_8 + \alpha_{10})\right)\sum_{i=1}^m \mathbb{E}[\|\hat{x}_i^{t-1}\|^2]
$$

$$
+ \left(\alpha_6 + \rho^{2M}K(\alpha_7 + \alpha_{11})\right)\sum_{i=1}^m \left(\mathbb{E}[\|\hat{y}_{1,i}^{t-1}\|^2] + \mathbb{E}[\|\hat{y}_{2,i}^{t-1}\|^2]\right) + (\alpha_7 + K\alpha_{14} + K\alpha_8)\frac{\sigma^2}{|\mathcal{B}|}. \tag{78}
$$

By choosing sufficiently large $K$ and $M$ satisfying

$$
K \geq \max\left\{\frac{2(92\alpha_4 + 91\alpha_3)}{91}, \frac{2(92\alpha_5 + 91\alpha_6)}{91}\right\},
$$

$$
M \geq \max\left\{\frac{\ln(1/2) - \ln(\alpha_8 + 2\alpha_{10})}{2\ln(\rho)}, \frac{\ln(1/2) - \ln(\alpha_7 + \alpha_{11})}{2\ln(\rho)},\right. \tag{79}
$$

$$
\left.\frac{\ln(1 - \alpha_1) - \ln(2) - \ln(\alpha_9 K)}{2\ln(\rho)}, \frac{\ln(1 - \alpha_2) - \ln(2) - \ln((\alpha_9 + \alpha_{12})K)}{2\ln(\rho)}\right\} \triangleq M_0,
$$

the following inequalities always hold:

$$\alpha_1 + \rho^{2M}\alpha_9 K \le \frac{1+\alpha_1}{2} < 1,$$

$$\alpha_2 + \rho^{2M}K(\alpha_9 + \alpha_{12}) \le \frac{1+\alpha_2}{2} < 1,$$

$$\alpha_4 + \rho^{2M}K(\alpha_8 + \alpha_{10}) < \frac{91}{92}\left(K - \alpha_3 - \rho^{2M}\alpha_{10}K\right), \tag{80}$$

$$\alpha_6 + \rho^{2M}K(\alpha_7 + \alpha_{11}) < \frac{91}{92}(K - \alpha_5).$$

Define an auxiliary function $V(t+1)$ as follows:

$$V(t+1) = \mathbb{E}[\|\overline{x}^{t+1} - x^*\|^2 + \|\overline{x}^{t+1} - \overline{x}^t\|^2] + \left(2 + \frac{125\beta}{31}\right)\mathbb{E}[\overline{\eta}^t(f(\overline{x}^t) - f(x^*))]$$

$$+ (K - \alpha_3 - \rho^{2M}\alpha_{10}K)\sum_{i=1}^{m}\mathbb{E}[\|\hat{x}_i^t\|^2] + (K - \alpha_5)\left(\sum_{i=1}^{m}\mathbb{E}[\|\hat{y}_{1,i}^t\|^2] + \mathbb{E}[\|\hat{y}_{2,i}^{t-1}\|^2]\right). \tag{81}$$

Set $\gamma = \max\{1 - \frac{\mu}{4L}, \frac{91}{92}\}$. Since $\alpha_1 = 1 - \frac{\mu}{2L}$, $\alpha_2 = \min\left\{\frac{45}{46}, \frac{1-r}{4\beta^2}\right\}$, and $\frac{1-r}{4\beta^2} < \frac{1}{4}$, we have

$$\max\left\{\frac{1+\alpha_1}{2}, \frac{1+\alpha_2}{2}, \frac{91}{92}\right\} \le \max\left\{1 - \frac{\mu}{4L}, \frac{91}{92}\right\} = \gamma. \tag{82}$$

It follows from equation 80 that

$$V(t+1) \le \gamma V(t) + (\alpha_7 + K\alpha_{14} + K\alpha_8)\frac{\sigma^2}{|\mathcal{B}|},$$

which is equivalent to

$$\left(V(t+1) - \frac{(\alpha_7 + K\alpha_{14} + K\alpha_8)\sigma^2}{(1-\gamma)|\mathcal{B}|}\right) \le \gamma\left(V(t) - \frac{(\alpha_7 + K\alpha_{14} + K\alpha_8)\sigma^2}{(1-\gamma)|\mathcal{B}|}\right). \tag{83}$$

Therefore, by using equation 83, we arrive at

$$V(t) \le \gamma^t V(0) + \frac{(\alpha_7 + K\alpha_{14} + K\alpha_8)\sigma^2}{(1-\gamma)|\mathcal{B}|}. \tag{84}$$

Moreover, since the relations $\overline{\eta}^{-1} = 0$, $\sum_{i=1}^{m}\|\hat{x}_i^{-1}\|^2 = 0$ and $\sum_{i=1}^{m}\|\hat{y}_{1,i}^{-1}\|^2 = 0$ hold, we have $V(0) = \|\overline{x}^0 - x^*\|^2 + \|\overline{x}^0\|^2$.

Furthermore, according to the definition of $V(t)$ in equation 81, we arrive at

$$\mathbb{E}[\|x_i^t - x^*\|^2] = \mathbb{E}[\|x_i^t - \overline{x}^t + \overline{x}^t - x^*\|^2] \le 2\mathbb{E}[\|\hat{x}_i^t\|^2] + 2\mathbb{E}[\|\overline{x}^t - x^*\|^2]$$

$$\le \max\left\{\frac{2}{K_1 - \alpha_3 - \rho^{2M_2}\alpha_{10}K_1}, 2\right\}V(t)$$

$$\le \max\left\{\frac{2V(0)}{K_1 - \alpha_3 - \rho^{2M_2}\alpha_{10}K_1}, 2V(0)\right\}\gamma^t \tag{85}$$

$$+ \left(\frac{(\alpha_7 + K\alpha_{14} + K\alpha_8)}{1-\gamma}\max\left\{\frac{2}{K_1 - \alpha_3 - \rho^{2M_2}\alpha_{10}K_1}, 2\right\}\right)\frac{\sigma^2}{|\mathcal{B}|},$$

which implies $\mathbb{E}\left[\|x_i^t - x^*\|^2\right] \le \mathcal{O}\left(\gamma^t\right) + \mathcal{O}\left(\frac{\sigma^2}{|\mathcal{B}|}\right)$ and Theorem 1.

### B.3 PROOF OF THEOREM 2

When accurate gradients are accessible to agents, Algorithm 1 reduces to the following algorithm.

---

**Algorithm 2** Deterministic version of Algorithm 1 (from agent $i$'s perspective)

---

1: **Input:** $x_i^0 \in \mathbb{R}^n$, $y_i^0 = \nabla f_i(x_i^0)$, $\eta_i^0 > 0$, $\beta \in (0, 1.36)$, $r \in (0, 1)$, $M \in \mathbb{N}^+$, and $T \in \mathbb{N}^+$.
2: **for** $t = 0, 1, \ldots, T$ **do**
3: $\quad x_i^{t+1}(0) = x_i^t - \eta_i^t y_i^t$
4: $\quad$ **for** $q = 0, 1, \ldots, M - 1$ **do**
5: $\quad\quad x_i^{t+1}(q+1) = \sum_{j \in \mathcal{N}_i} w_{ij} x_j^{t+1}(q)$
6: $\quad$ **end for**
7: $\quad x_i^{t+1} = x_i^{t+1}(M)$
8: $\quad y_i^{t+1}(0) = y_i^t + \nabla f_i(x_i^{t+1}) - \nabla f_i(x_i^t)$
9: $\quad$ **for** $q = 0, 1, \ldots, M - 1$ **do**
10: $\quad\quad y_i^{t+1}(q+1) = \sum_{j \in \mathcal{N}_i} w_{ij} y_j^{t+1}(q)$
11: $\quad$ **end for**
12: $\quad y_i^{t+1} = y_i^{t+1}(M)$
13: $\quad L_i^{t+1} = \frac{\|y_{i,1}^{t+1} - y_{i,2}^t\|}{\|x_i^{t+1} - x_i^t\|}$ if $x_i^{t+1} \neq x_i^t$; otherwise, $L_i^{t+1} = 1$
14: $\quad \eta_i^{t+1} = \min\left\{ \beta \eta_i^t, \frac{7\sqrt{r}}{10} \frac{\eta_i^t}{\sqrt{[(\eta_i^t L_i^{t+1})^2 - 1]_+}} \right\}$
15: **end for**

---

*Proof of Theorem 2:* By using an argument similar to the derivation of equation 44, we obtain

$$\|\overline{x}^{t+1} - x^*\|^2 + \|\overline{x}^{t+1} - \overline{x}^t\|^2 + \left(2 + \frac{125\beta}{31}\right)\overline{\eta}^t f(\overline{x}^t) - f(x^*)$$

$$\leq (1 - \mu\eta_{\min} + a_9 \mu \eta_{\max})\|\overline{x}^t - x^*\|^2 + \frac{45}{46}\|\overline{x}^t - \overline{x}^{t-1}\|^2$$

$$+ \gamma\left(2 + \frac{125\beta}{31}\right)\overline{\eta}^{t-1}(f(\overline{x}^{t-1}) - f(x^*)) + \delta_5^t, \tag{86}$$

for $\gamma \in (0, 1)$, where the constant and $\delta_5^t$ is given by

$$\delta_5^t = \frac{125}{62m}\left(\frac{L^2\eta_{\max}}{a_6} + \frac{2\beta^2}{a_8} + b_1 + \frac{49(1+a_7)}{50a_7} + \frac{124L^2\eta_{\max}}{125a_9\mu}\right)\sum_{i=1}^m \|\hat{x}_i^t\|^2$$

$$+ \frac{125}{62m}\left(\frac{2\beta^2}{a_8} + b_1 + \frac{49(1+a_7)}{50a_7}\right)\sum_{i=1}^m \|\hat{x}_i^{t-1}\|^2$$

$$+ \frac{\eta_{\max}^2}{m}\left(\frac{187}{31} + \frac{125}{31a_2} + \frac{2}{a_1} + \frac{2}{a_9\mu\eta_{\max}}\right)\sum_{i=1}^m \|\hat{y}_i^t\|^2$$

$$+ \frac{125\eta_{\max}^2}{62m}\left(\frac{1}{a_3} + 1 + \frac{1}{a_8} + \frac{2}{a_2}\right)\sum_{i=1}^m \|\hat{y}_i^{t-1}\|^2. \tag{87}$$

By using an argument similar to the derivations of equation 54 and equation 55, we have

$$\sum_{i=1}^m \|\hat{x}_i^t\|^2 \leq \rho^{2M}\left(12\eta_{\max}^2 \sum_{i=1}^m \|\hat{y}_i^{t-1}\|^2 + (24\eta_{\max}^2 L^2 + 3)\sum_{i=1}^m \|\hat{x}_i^{t-1}\|^2\right.$$

$$\left. + 48m\eta_{\max}^2 L^2 \|\overline{x}^t - x^*\|^2 + 48m\eta_{\max}^2 L^2 \|\overline{x}^t - \overline{x}^{t-1}\|^2\right), \tag{88}$$

$$\sum_{i=1}^m \|\hat{y}_{1,i}^t\|^2 \leq \rho^{2M}\left(18L^2 \sum_{i=1}^m \|\hat{x}_i^t\|^2 + 18L^2 \sum_{i=1}^m \|\hat{x}_i^{t-1}\|^2 + 3\sum_{i=1}^m \|\hat{y}_{1,i}^{t-1}\|^2\right.$$

$$\left. + 18mL^2 \|\overline{x}^t - \overline{x}^{t-1}\|^2\right). \tag{89}$$

By using an argument similar to the derivation of equation 84 and constructing the following function:

$$V(t+1) = \|\overline{x}^{t+1} - x^*\|^2 + \|\overline{x}^{t+1} - \overline{x}^t\|^2 + \left(2 + \frac{125\beta}{31}\right)\overline{\eta}^t(f(\overline{x}^t) - f(x^*))$$
$$+ (K - \alpha_3 - \rho^{2M}\alpha_{10}K)\sum_{i=1}^{m}\|\hat{x}_i^t\|^2 + (K - \alpha_5)\sum_{i=1}^{m}\|\hat{y}_i^t\|^2, \tag{90}$$

we obtain the following relationship:

$$V(t+1) \leq \gamma V(t), \tag{91}$$

which implies $V(t) \leq \gamma^t V(0)$. Then, following an argument similar to the derivations of equation 85, we arrive at $\|x_i^t - x^*\|^2 \leq \mathcal{O}\left(\gamma^t\right)$, which proves Theorem 2.

### B.4 PROOF OF COROLLARY 1

According to Theorem 1, the convergence rate of Algorithm 1 is $\mathcal{O}\left(\gamma^T\right) + \mathcal{O}\left(\frac{\delta^2}{|\mathcal{B}|}\right)$. Hence, to find an $\epsilon$-optimal solution, the number of outer-loop iterations $T$ needs to satisfy $T = \mathcal{O}(\log(\epsilon^{-1}))$. At each outer-loop iteration, Algorithm 1 requires $|\mathcal{B}|$ gradient evaluations at both $g_i^t(x_i^{t+1})$ and $g_i^t(x_i^t)$, resulting in a total of $2|\mathcal{B}|$ evaluations. Meanwhile, Lines 3, 8, and 9 in Algorithm 1 require $M$ gradient evaluations at $x_{i,1}^{t+1}(0)$, $y_{i,1}^{t+1}(0)$, and $y_{i,2}^t(0)$, Lines 5, 11, and 12 in Algorithm 1 require $M$ gradient evaluations at $x_i^{t+1}(q)$, $y_{i,1}^{t+1}(q)$, and $y_{i,2}^t(q)$; and Lines 15 and 16 in Algorithm 1 each require one gradient evaluation at $L_i^{t+1}$ and $\eta_i^{t+1}$, respectively. Based on the above discussion, we have that Algorithm 1 requires at most $2|\mathcal{B}| + 3M + 3$ gradient evaluations per outer-loop iteration $t$, leading to a computational complexity of $\mathcal{O}((2|\mathcal{B}| + 3M + 3)\log(\epsilon^{-1}))$ over $T$ iterations. In the deterministic setting, Algorithm 1 reduces to Algorithm 2, which requires at most $2M + 3$ gradient evaluations per outer-loop iteration $t$, and thus has a computational complexity of $\mathcal{O}((2M + 3)\log(\epsilon^{-1}))$ over $T$ iterations.

## C EXPERIMENTAL SETUPS AND ADDITIONAL EXPERIMENTAL RESULTS

### C.1 BENCHMARK DATASETS

**MNIST.** The "MNIST" dataset is a benchmark dataset widely used in machine learning and computer vision (Deng, 2012). It typically consists of $70,000$ grayscale images of handwritten digits (i.e., 0–9), with $60,000$ used for training and $10,000$ for testing. Each image has a size of $28 \times 28$ pixels, with the digit centered in the frame.

**CIFAR-10.** The "CIFAR-10" dataset consists of $60,000$ color images of size $32 \times 32$ pixels in 10 classes, with $6,000$ images per class (Krizhevsky et al., 2010). Among them, $50,000$ images are used for training and $10,000$ for testing. The dataset covers a diverse set of object categories, including airplanes, automobiles, birds, cats, deer, dogs, frogs, horses, ships, and trucks. Compared with the "MNIST" dataset, the "CIFAR-10" dataset poses a greater challenge due to its colored and natural images with larger intra-class variability.

**CIFAR-100.** The "CIFAR-100" dataset is a natural extension of the "CIFAR-10" dataset (DeVries & Taylor, 2017). It contains $60,000$ color images of size $32 \times 32$ pixels, and spreads across 100 classes with 600 images per class. The $50,000$ images are used for training and $10,000$ for testing. However, due to its larger number of categories and the fine-grained nature of many classes, the "CIFAR-100" dataset is regarded as the most challenging dataset within the CIFAR series.

**Mushrooms.** The "Mushrooms" dataset is a classic benchmark dataset from the UCI Machine Learning Repository (Tutuncu et al., 2022). It contains $8,124$ instances of gilled mushrooms, each described by 22 categorical attributes, such as cap shape, surface, and color. The prediction task is to classify each mushroom as either edible or poisonous. In this paper, we focus on $l_2$-logistic regression on the "Mushrooms" dataset, as the task naturally fits into a binary classification problem.

**Shakespeare.** The "Shakespeare" dataset contains $3,829,611$ training samples and $1,646,425$ test samples. Each sample consists of a sequence of 80 characters and the subsequent character to be

predicted. The dataset is derived from the lines of various characters in Shakespeare's plays. Due to the diversity of characters and scenes, the next character to appear often varies significantly. This dataset is regarded as a highly challenging benchmark.

## C.2 EXPERIMENTAL SETUPS

**Convolutional neural network (CNN) training.** For the "MNIST" dataset, we trained a two-layer CNN. The first convolutional layer has 64 output channels with $3 \times 3$ kernels, stride 1, and padding 1, followed by batch normalization, LeakyReLU activation, and $2 \times 2$ max pooling. The second convolutional layer has 128 output channels with the same kernel configuration. The feature maps are then passed through adaptive average pooling to a $1 \times 1$ representation, flattened, and fed into a fully connected layer to produce the output classes. The model was trained with a batch size of 128 using the cross-entropy loss.

For the "CIFAR-10" dataset, we trained a four-layer CNN consisting of four convolutional layers with progressively increasing channel sizes of 32, 64, 128, and 256. Each convolution uses a $3 \times 3$ kernel with stride 1 and padding 1. To stabilize training and reduce spatial resolution, we employed batch normalization, a LeakyReLU activation, and $2 \times 2$ max pooling after every convolutional block. The resulting feature maps are aggregated by adaptive average pooling to a $1 \times 1$ representation, which is then flattened and passed to a fully connected layer to produce the final class predictions. The model was trained with a batch size of 128 using the cross-entropy loss.

For the "CIFAR-100" dataset, we trained a five-layer CNN with residual connections to enhance feature extraction. The network begins with a 32-channel convolutional layer ($3 \times 3$ kernels, stride 1, padding 1), followed by batch normalization, LeakyReLU activation, and $2 \times 2$ max pooling. The subsequent convolutional blocks progressively increase the channels to 64, 128, 256, and 512. To enhance feature extraction, we introduced residual paths: one from the raw input through a $2 \times 2$ convolution with stride 2, another from the second block via a $2 \times 2$ convolution, and a direct path from the raw input via an $8 \times 8$ convolution. The model was trained with a batch size of 128 using the cross-entropy loss.

**Logistic regression.** For the logistic regression task using the "Mushrooms" dataset, we employed a single-layer linear model, which directly maps the 22 input features to two output logits corresponding to the classes. Training was conducted using the loss function given in equation 92.

**Recurrent neural network (RNN) training.** For the "Shakespeare" dataset, we trained an LSTM-based recurrent neural network. The model first maps each input token to a dense vector through an embedding layer with an embedding dimension of 8. The embedded sequence is then fed into a single-layer LSTM with a hidden size of 128 and batch-first input formatting. Finally, the representation is passed through a fully connected layer to project it back to the vocabulary space, producing logits for the next-character prediction.

## C.3 ADDITIONAL EXPERIMENTAL RESULTS

In this section, we provide five additional experimental results: (1) the performance evaluation of Algorithm 1 on logistic regression with strongly convex and smooth loss functions; (2) the performance evaluation of Algorithm 1 on next-characterize prediction tasks using the "Shakespeare" dataset; (3) the comparison of Algorithm 1 and distributed ADAM in Nazari et al. (2022); (4) the performance evaluation of Algorithm 1 under different $\beta$, $r$, and $M$, respectively, on the "MNIST" dataset; and (5) the performance evaluation of Algorithm 1 under various network topologies.

**(1) Logistic regression using the "Mushrooms" dataset.** We evaluate the effectiveness of Algorithm 1 by using an $l_2$-logistic regression classification problem on the "Mushrooms" dataset (Tutuncu et al., 2022). To ensure heterogeneous data distribution, we spread data samples among five agents according to their target values. Specifically, agents 1, 2, and 3 have samples with the target value of 0, while agents 4 and 5 have samples with the target value of 1. All agents cooperatively learn an optimal model parameter $x^*$ to problem 1, in which the loss function of agent $i$ is given by

$$l(x, \xi_i) = \frac{1}{|\mathcal{B}|} \sum_{j=1}^{|\mathcal{B}|} \left( -(1 - b_{ij}) \ln \left( \frac{e^{x_1 a_{ij}}}{e^{x_1 a_{ij}} + e^{x_2 a_{ij}}} \right) - b_{ij} \ln \left( \frac{e^{x_2 a_{ij}}}{e^{x_1 a_{ij}} + e^{x_2 a_{ij}}} \right) + \frac{L_2}{2} \|x\|^2 \right),$$

(92)

where $|\mathcal{B}|$ represents the number of sampled data points per iteration. In this experiment, we used a full batch setting, i.e., $|\mathcal{B}| = |\mathcal{D}_i|$ with $\mathcal{D}_i$ denoting the local dataset of agent $i$. Here, $x = [x_1, x_2]^\top$ is the model parameter and the positive constant $L_2$ is a regularization parameter. It is clear that the loss function in equation 92 is strongly convex and smooth.

In this experiment, we compared the test accuracies of Algorithm 1 with existing distributed optimization algorithms, including distributed GD in Nedic & Ozdaglar (2009) and deterministic GT in Nedić et al. (2017). The stepsizes for distributed GD and deterministic GT are the same as those employed in our "MNIST" experiment in the main text (i.e., $\eta_i^t = \frac{0.1}{(1+t)^{0.5}}$ for distributed GD and $\eta_i = 0.1$ for deterministic GT). The training process spanned 250 iterations.

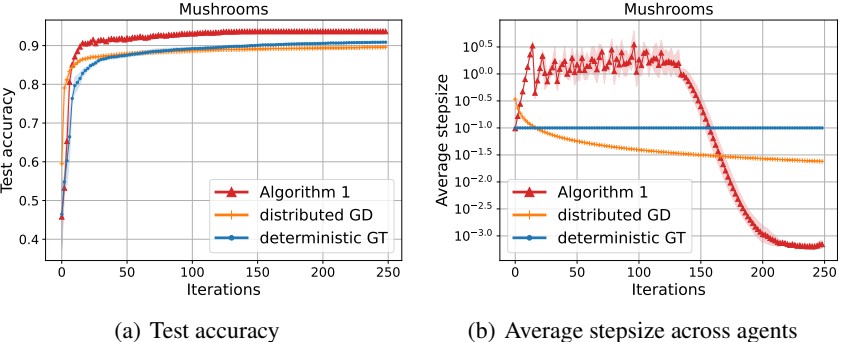

(a) Test accuracy          (b) Average stepsize across agents

Figure 6: Test-accuracy and average-stepsize (across five agents) evolutions of Algorithm 1, distributed GD in Nedic & Ozdaglar (2009), and deterministic GT in Nedić et al. (2017). The 95% confidence intervals were computed from three independent runs with seeds 42, 1010, and 2024.

Fig. 6(a) shows that Algorithm 1 achieves the highest test accuracy and convergence speed compared with distributed GD and deterministic GT. This is because larger stepsizes is allowed in the early stages of Algorithm 1 than distributed GD and deterministic GT (as shown in Fig. 6(b)). Furthermore, Fig. 6 shows that Algorithm 1 exhibits stable convergence accuracy after 200 iterations. This result implies a clear stopping criterion for our algorithm, that is, by setting $\tau = 10^{-3}$, each agent $i$ can stop training once $|\eta_i^t| < \tau$.

**(2) Next-characterize prediction using the "Shakespeare" dataset.** We evaluate the learning accuracy of Algorithm 1 using a next-characterize prediction task on the "Shakespeare" dataset. To ensure heterogeneous data distribution, we spread data samples among five agents according to a Dirichlet distribution with parameter $\alpha = 0.5$.

In this experiment, we compared the test accuracies of Algorithm 1 with existing distributed stochastic optimization algorithms, including distributed SGD in Jakovetic et al. (2018) and stochastic GT in Pu & Nedić (2021). The stepsize for distributed SGD was set to $\eta_{i,t} = \frac{10}{(1+t)^{0.51}}$ while the stepsize for stochastic GT was set to $\eta = 0.5$. The training process spanned 200 epochs.

Fig. 7(a) shows that Algorithm 1 outperforms both distributed SGD and stochastic GT in test accuracy. This improvement can be attributed to the larger stepsizes allowed by our adaptive stepsize approach, as evidenced by Fig. 7(b).

**(3) Comparison of Algorithm 1 and distributed ADAM in Nazari et al. (2022).** To compare the convergence accuracy of Algorithm 1 with the existing adaptive stepsize approach for distributed (online) learning, i.e., distributed ADAM in Nazari et al. (2022), we conducted additional experiments by comparing their test accuracies on image classification using the "CIFAR-10" dataset.

Fig. 8(a) shows that our Algorithm 1 outperforms distributed ADAM in terms of both test accuracy and steady-state performance. Furthermore, Fig. 8(b) indicates that the stepsize in distributed ADAM decays rapidly, which leads to a low convergence speed in the later stages of the algorithm.

**(4) The effects of $\beta$, $r$, and $M$ on convergence accuracy with respect to the "MNIST" dataset.** We evaluate the test accuracies of Algorithm 1 under different coefficients $\beta$ and $r$ in the stepsize

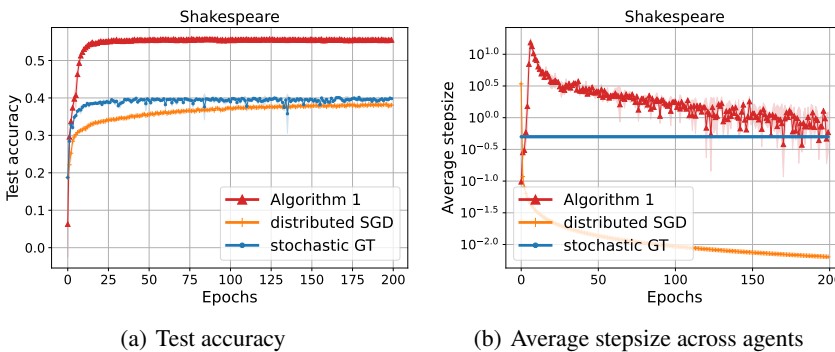

(a) Test accuracy

(b) Average stepsize across agents

Figure 7: Test-accuracy and average-stepsize (across ten agents) evolutions of Algorithm 1, distributed SGD in Jakovetic et al. (2018), and stochastic GT in Pu & Nedić (2021). The 95% confidence intervals were computed from three independent runs with seeds 42, 1010, and 2024.

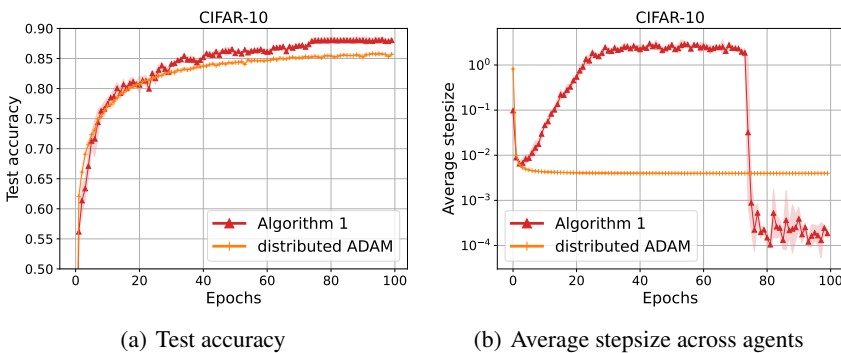

(a) Test accuracy

(b) Average stepsize across agents

Figure 8: Test-accuracy and average-stepsize (across five agents) evolutions of Algorithm 1 and distributed ADAM (Nazari et al., 2022). The 95% confidence intervals were computed from three independent runs with random seeds 42, 1010, and 2024.

update rule (i.e., Line 16 in Algorithm 1) and the number of inner-loop iterations $M$ in Algorithm 1, respectively. We ran this experiment on the "MNIST" dataset over 20 epochs, with a batch size of 128 and a random seed as 42.

Fig. 9(a), Fig. 9(b), Fig. 9(d), and Fig. 9(e) imply that larger $\beta$ and $r$ lead to faster convergence and earlier stopping in Algorithm 1. This result is intuitively consistent, as large $\beta$ and $r$ contribute to larger stepsizes before convergence stages (as shown in Fig. 9(d) and Fig. 9(e)), which in turn leads to a higher convergence speed. Furthermore, Fig. 9(c) and Fig. 9(f) show that the number of inner-consensus-loop iterations $M$ has a negligible effect on convergence accuracy and the stopping criterion. Hence, in practical machine learning tasks, we can set $M = 1$ (so that Algorithm 1 reduces to a single-loop algorithm) to minimize the communication cost of Algorithm 1. The experimental results in Fig. 9 further confirm the default parameter configuration $(\beta, r, M) = (1.35, 0.99, 1)$ for our algorithm, which align with the discussion in the subsection "The effects of $\beta$, $r$, and $M$ on convergence accuracy" (with respect to the "CIFAR-10" dataset) in the main text.

**(5) Performance evaluation of Algorithm 1 under various network topologies.** We conducted experiments to evaluate the efficacy of our Algorithm 1 under different network topologies. We considered a network of $m = 10$ agents, with the interaction graph being a ring network and random $d$-regular graph Bollobás (1986) with $d$ (called "Degree" in Fig. 10) set to 2, 3, 5, and 8. We used the same parameters as those employed in subsection "Comparison with existing distributed stochastic optimization approaches" in our main text. The experimental results are shown in Fig. 10.

The experimental results in Fig. 10(a) and Fig. 10(b) show that the impact of network topologies on the convergence accuracy of our algorithms is slight when Assumption 2 is satisfied.

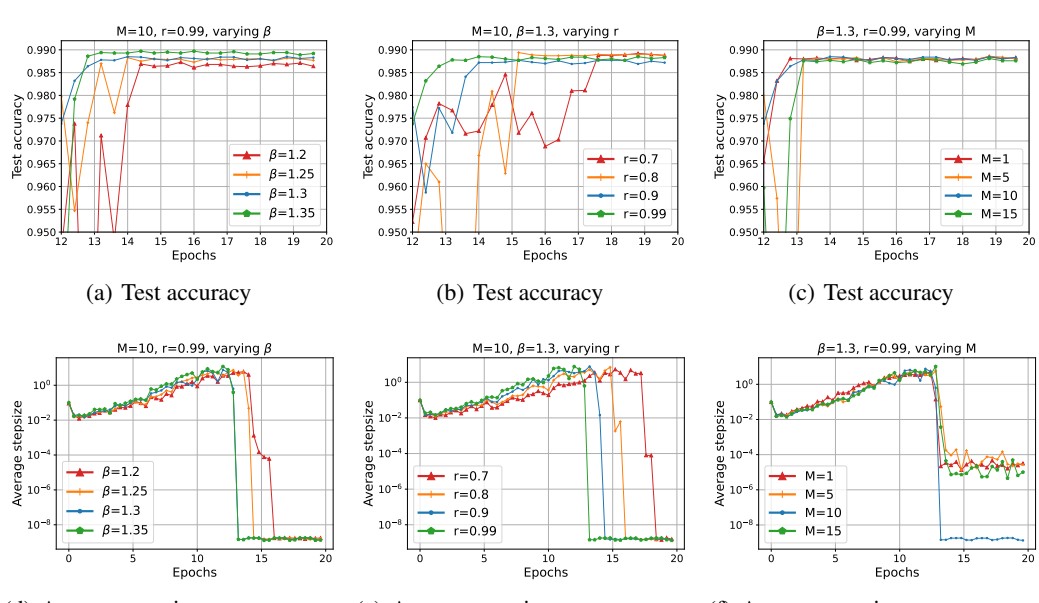

(a) Test accuracy      (b) Test accuracy      (c) Test accuracy

(d) Average stepsize across agents   (e) Average stepsize across agents   (f) Average stepsize across agents

Figure 9: Test-accuracy and average-stepsize (across five agents) evolutions of Algorithm 1 under different parameters $\beta$, $r$, and $M$, respectively.

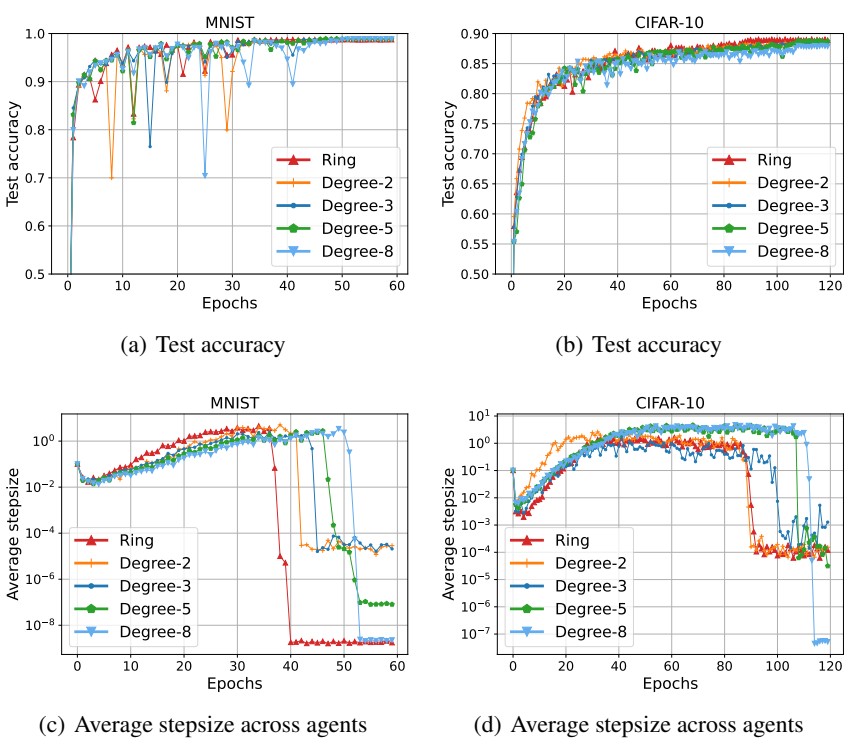

(a) Test accuracy         (b) Test accuracy

(c) Average stepsize across agents    (d) Average stepsize across agents

Figure 10: Test-accuracy and average-stepsize (across ten agents) evolutions of Algorithm 1 under different network topologies.

