# OpenReview forum: "Eliminating Steady-State Oscillations in Distributed Optimization and Learning via Adaptive Stepsize"
_ICLR.cc/2026/Conference — Submitted to ICLR 2026_

### Official Review · Reviewer_WeYr · 2025-10-16

**Soundness:** 1
**Presentation:** 1
**Contribution:** 1
**Rating:** 0
**Confidence:** 5

**Summary:**

The paper studies adaptive algorithms in federated learning where there is no central server involved in the stepsize scheduling. They call this a fully decentralized setting. Then, they provided an adaptive stepsize algorithm which is claimed to eliminate steady-state oscillations. Numerical experiments are performed to validate the results.

**Strengths:**

The fully decentralized stepsize scheduling seems to be a new setting. In this setting, the authors provided a method and showed linear convergence under some circumstances.

**Weaknesses:**

First of all, I find the motivation of this setting quite weak. In the stepsize scheduling, the number of bits to communicate is usually a small constant (compared to communicating a full gradient). Therefore, even in a decentralized network, this usually takes much less time than the gradient gossipings. I cannot see any practical reasons why the (global) stepsize coordination takes too much of communication time.

Moreover, the theorem statements in the paper did not provide good convergence guarantees. For instance, in eq. (2) in Theorem 1, this is not even a convergence result with a non-vanishing term of $O (\frac {\sigma^2} {|\mathcal B|})$.

I feel very suspicious about the correctness of the main technical results. For instance, in Corollary 1, the authors claimed to get an $\epsilon$-solution in $O ((|\mathcal B| + M) \log \frac 1 \epsilon)$ gradients. This claim also does not follow from Theorem 1.

**Questions:**

See the “weakness” paragraphs above. I believe that the setting of fully decentralized stepsize scheduling is not well motivated and the main technical claims in the paper are wrong.

---

> ### Author Response · Authors · 2025-11-23
> **Response to Reviewer WeYr**
>
> ${\color{blue} \text{Response:}}$ We respectfully note that the reviewer has misunderstand the motivation and the core idea of our work.
>
> ${\color{blue} \text{1. Regarding the comment of directly transmitting the stepsize between agents}}$
>
> The key enabler of our approach to achieve an improved convergence performance is that we allow agents to use $\color{black}\text{heterogeneous and adaptive stepsizes}$, which is the opposite of the statement made by the reviewer.
>
> Specifically, if the stepsize is directly transmitted or synchronized among all agents—as suggested by the reviewer—then all agents end up using a single global stepsize. This essentially reduces our approach to a standard uniform-stepsize scheme and, consequently, removes the benefits of heterogeneous and adaptive stepsizes in terms of convergence performance, including fast early-stage convergence and the elimination of steady-state oscillations, as demonstrated in Fig. 1–3.
>
> ${\color{blue} \text{2. Regarding the comment of the convergence guarantees in Theorem 1}}$
>
> It is well known that distributed stochastic optimization without diminishing stepsizes or variance-reduction techniques inherently leads to a non-vanishing error bound of order $\mathcal{O}(\frac{\sigma^2}{|\mathcal{B}|})$. This term is caused by the finite batch size and is an intrinsic limitation of all stochastic optimization methods that rely on finite samples (see, e.g., Pu et al., 2021; Koloskova et al., 2021). Therefore, the presence of this term in Theorem 1 is expected and fully consistent with standard results in the literature.
>
> ${\color{blue} \text{3. Regarding the comment of the correctness of Corollary 1 and its relation to Theorem 1}}$
>
> Thank you for the comment. In the initial version, Corollary 1 indeed overlooked the effect of the non-vanishing term $\mathcal{O}(\frac{\sigma^2}{|\mathcal{B}|})$ on computational complexity. In the revised manuscript, we have revised Corollary 1 as follows:
>
> Corollary 1. Under Assumptions 1 and 2, for any $\epsilon>0$, Algorithm 1 with noisy gradient estimates requires at most $\mathcal{O}((2|\mathcal{B}|+3M+3)\log(\epsilon^{-1}))$ gradient evaluation to obtain an $\epsilon+\mathcal{O}(\frac{\sigma^2}{|\mathcal{B}|})$-solution, and Algorithm 1 with accurate gradients requires at most $\mathcal{O}((2M+3)\log(\epsilon^{-1}))$ gradient evaluation to obtain an $\epsilon$-solution.
>
> [1] Shi Pu and Angelia Nedic. Distributed stochastic gradient tracking methods.  Mathematical Pro-gramming, 187(1):409–457, 2021.
>
> [2] Anastasiia Koloskova, Tao Lin, and Sebastian U Stich. An improved analysis of gradient tracking
> for decentralized machine learning. Advances in Neural Information Processing Systems, 34:
> 11422–11435, 2021.

---

> ### Comment · Reviewer_WeYr · 2025-11-23
>
> I thank the authors for the response and for addressing my concerns on the motivation of the paper.
> Moreover, the mistake in Corollary 1 is fixed. In light of these, I will increase the score from 0 to 4.
>
> Now the derivation from Theorem 1 to Corollary 1 is correct. But, I believe that the Theorem 1 and 2 are still problematic. This is also noted by another reviewer, as the correct or optimal stepsize choices should ensure scaling invariance.
>
> Moreover, in the rebuttal, the authors mentioned that the $\frac {\sigma^2} B$ term is inevitable and cited some papers as evidence. However, the authors' statement is in fact incorrect. Please just check the Theorem 2 in the reference [2] you cited. The variance term in their analysis is divided by the iteration $T$, while in your result it is not the case.
>
> I hope the authors would kindly understand that my updated score of 4 is already a bit high for the current presentation. And I hope they will fix the issue in their main theorem carefully in the later revision.

---

### Official Review · Reviewer_WeSj · 2025-10-28

**Soundness:** 2
**Presentation:** 3
**Contribution:** 2
**Rating:** 4
**Confidence:** 4

**Summary:**

This work solves the problem of stepsize selection in distribution optimization. For this purpose, the authors propose Algorithm which automatically sets the value of stepsize and prove the convergence guarantee for this algorithm. Algorithm 1 achieves comparable convergence rate with the centralized optimization. Experimental results confirm the effectiveness of Algorithm 1.

**Strengths:**

1. This work studies an important problem of selecting stepsize to avoid both slow convergence and steady-state oscillations.

2. The authors provide stong theoretical guarantee for the proposed algorithm.

3. The experiment results (such as Figure 1) demonstrate the claims of this work.

**Weaknesses:**

1. Hyperparameter tuning: While automatically setting the stepsize, Algorithm 1 introduces additional parameters including $\beta$ $r$ and $M$, requiring futher tuning. The experiments in Figure 3 cannot solve this problem, as training with a batch size of 128 on the MNIST dataset is a very stable process

2. Lack of large-scale experiments: The experiments use five nodes and a specific data distribution, which may be not practical. It is hard to see if the results can generalize to other setups.

3. If I understand correctly, Theorem 1 requires $M > M_0$  inner-consensus-loop iterations for convergence. It is hard to estimate the value of $M_0$. If this value is very large, the theoretical gurantee does not hold in practice.

**Questions:**

1. How to select the values of $\beta$ $r$ and $M$ in applications?

2. How do Algorithm 1 compare with baselines in a large-scale setup and different data distributions?

3. How to select coupling weight W? Is the choice in Section 6 manually tuned or randonly set?

4. In Figure 1(f), why does the Stepsize decrease suddenly? Is this phenomenon met in every running of CIFAR10?

---

> ### Author Response · Authors · 2025-11-26
> **Response to Reviewer WeSj: Part I**
>
> ${\color{orange} \text{Response to Weakness 1: }}$
>
> We thank the reviewer for the time spent reviewing our manuscript. We emphasize that our goal is to
> develop an adaptive stepsize approach that can enhance convergence performance in distributed learning---including eliminating steady-state oscillations and ensuring fast convergence---rather than to achieve fully automatic stepsize selection. To this end, we introduce the parameters $\beta$, $r$, and $M$, which assists the agents during stepsize adjustment.
>
> Following the reviewer's suggestion, we have conducted additional experiments on the "CIFAR-10" dataset to evaluate the convergence performance of our algorithm under different $\beta$, $r$, and $M$, respectively. The experimental results are summarized in Fig. 3 in the revised manuscript. (Note that we have moved the experimental results on the test accuracies of Algorithm 1 under different $\beta$, $r$, and $M$ on the ``MNIST" dataset to Fig. 9 in Appendix C.3 on page 36.)
>
> The updated Fig. 3 and Fig. 9 consistently show that the parameter configuration $\beta = 1.35$, $r = 0.99$, and $M = 1$ provides the best performance for Algorithm 1 across both datasets. These results suggest a default parameter configuration $(\beta,r,M)=(1.35,0.99,1)$ for Algorithm 1, which helps ease the tuning effort in real-world applications.}
>
> ${\color{orange} \text{Response to Weakness 2: }}$
>
> Thank you for the constructive comment. Following the reviewer's suggestion, in the revised manuscript, we have added two new subsections "The effect of network size $m$ on convergence accuracy" and "The effect of data heterogeneity across agents on convergence accuracy" to Section 6. These additions evaluate the performance of Algorithm 1 under varying network sizes and under different data distributions across agents, respectively.
>
> Specifically, we first conducted additional experiments on the "CIFAR-10" dataset to compare test accuracies of Algorithm 1 with two baseline algorithms---distributed SGD in Jakovetic et al. (2018) and stochastic GT in Pu \& Nedi\'c (2021)---under different network sizes $m=10$, $m=15$, $m=20$, respectively. The experimental results, summarized in Fig. 4 of the revised manuscript (page 10), show that Algorithm 1 consistently achieves higher test accuracy and more stable steady-state convergence than both distributed SGD and stochastic GT, regardless of the network size $m$.
>
> Furthermore, we also conducted additional experiments on the "CIFAR-10'' dataset to compare the test accuracies of Algorithm 1 with those of distributed SGD and stochastic GT under different levels of data heterogeneity across agents. Specifically, we partitioned the "CIFAR-10'' dataset among five agents using the Dirichlet partitioning scheme with parameters $\alpha = 0.1$, $\alpha = 0.5$, and $\alpha = 10$, respectively (note that a smaller $\alpha$ corresponds to a higher level of data heterogeneity among agents). The experimental results, summarized in Fig. 5 of the revised manuscript (page 10), show that Algorithm 1 maintains higher test accuracy and more stable steady-state convergence than both distributed SGD and stochastic GT across all levels of data heterogeneity.
>
> We hope that the newly added Fig. 4 and Fig. 5 can address the reviewer’s concerns regarding the applicability of our results to different network sizes and levels of data heterogeneity.}
>
> ${\color{orange} \text{Response to Weakness 3: }}$
>
> Thank you for the comment. We acknowledge that the condition $M>M_{0}$ is required to establish the convergence guarantee of our algorithm. Although the current convergence result is somewhat conservative, our work is, to the best of our knowledge, ${\color{black} \text{the first}}$ to demonstrate that adaptive stepsize approaches can both eliminate steady-state oscillations and ensure fast convergence in distributed learning.
>
> In addition, we explain that, historically, limited theory has never yet prevented practitioners from using methods in a broader setting. One notable example is ADAM, which has been shown to fail to converge to an optimal solution even in simple one-dimensional settings [1], however, it is still widely employed in a variety of machine learning tasks. In fact, our experimental results in Fig. 3(c) and Fig. 9(c) demonstrate that our approach also works well in various machine learning tasks with $M=1$, implying that its utility extends beyond the current theoretical results. Hence, we believe that our approach is effective in real-world applications for any $M\geq 1$.
>
> [1] Reddi S J, Kale S, Kumar S. On the convergence of ADAM and beyond.In International Conference on Learning Representations, 2018.

---

> ### Author Response · Authors · 2025-11-26
> **Response to Reviewer WeSj: Part II**
>
> ${\color{orange} \text{Response to Question 1:}}$
>
> As stated in our response to Weakness 2, we recommend using the parameter configuration (β, r, M ) = (1.35, 0.99, 1) for Algorithm 1 in real-world applications.
>
> ${\color{orange} \text{Response to Question 2:}}$
>
> In the revised manuscript, we have conducted additional experiment to evaluate the effectiveness of Algorithm 1 under varying network sizes and under different data distributions across agents, respectively. The experimental results are summarized in Fig. 4 and Fig. 5 in the newly added subsections "The effect of network size $m$ on convergence accuracy" and "The effect of data heterogeneity across agents on convergence accuracy" in Section 6 in the revised mansucript on page 9 to page 10.
>
> ${\color{orange} \text{Response to Question 3:}}$
>
>  We can choose any coupling matrix $W$ that satisfies Assumption 2. In Section 6, the weights of $W$ are randomly set, as long as it satisfies Assumption 2. For example, in a ring topology, we can choose $w_{ij}=0.2$ and $w_{ii}=0.6$, or $w_{ij}=0.1$ and $w_{ii}=0.8$.
>
> In addition, we conducted additional experiments to evaluate the effectiveness of our Algorithm 1 under different network topologies. The experimental results are summarized in the newly added Fig. 10 in Appendix C.3 in the revised manuscript on page 36.
>
> ${\color{orange} \text{Response to Question 4:}}$
>
> Thank you for the comment. The sudden decrease in the stepsize in Fig. 1(f) occurs because Algorithm 1 is approaching a stable solution to the image classification problem on the "CIFAR-100" dataset (see Fig. 1(c) for details). This phenomenon highlights a key advantage of our adaptive stepsize approach over existing counterparts---it allows agents to take large stepsizes during the early learning stages and automatically reduces the stepsizes as Algorithm 1 enters the convergence region. This advantage explains why our adaptive stepsize approach can eliminate steady-state oscillations and ensure fast convergence. In addition, we note that existing distributed optimization algorithms (e.g., distributed SGD with diminishing stepsizes in Jakovetic et al. (2018) and stochastic GT with fixed stepsizes in Pu \& Nedi\'c (2021)) and existing adaptive-stepsize approaches (e.g., ADAM in  Kingma (2014) and adaptive SGD in  Malitsky \& Mishchenko (2024)) do to share this advantage. As a result, they often exhibit slower convergence or larger steady-state oscillations, as demonstrated in Figs. 1(a)–1(c) and 2(a)–2(c).
>
> [1] Shi Pu and Angelia Nedic. Distributed stochastic gradient tracking methods. Mathematical Programming, 187(1):409–457, 2021.
>
> [2] Dusan Jakovetic, Dragana Bajovic, Anit Kumar Sahu, and Soummya Kar. Convergence rates for distributed stochastic optimization over random networks. In IEEE Conference on Decision and Control, pp. 4238–4245. IEEE, 2018.
>
> [3] Diederik P Kingma. Adam: A method for stochastic optimization. arXiv preprint arXiv:1412.6980, 2014.
>
> [4] Yura Malitsky and Konstantin Mishchenko. Adaptive proximal gradient method for convex optimization. Advances in Neural Information Processing Systems, 37:100670–100697, 2024.

---

> > ### Comment · Reviewer_WeSj · 2025-11-27
> >
> > Thank you for your response.
> > After reading the responses and other review comments, I decide to keep the same score. My concerns are:
> >
> > 1. The Theory-Practice Gap between the theoretical requirement (large inner-loops K) and the practical implementation (K=1)
> >
> > 2. Hyperparameter Selection: Three new hyperparameters essentially introduce addtional tuning complexity.
> >
> > 3. Theoretical Correctness, as mentioned by other reviewers.

---

### Official Review · Reviewer_snAG · 2025-10-29

**Soundness:** 3
**Presentation:** 4
**Contribution:** 3
**Rating:** 6
**Confidence:** 5

**Summary:**

This paper proposes an adaptive stepsize approach for distributed stochastic optimization and learning, which can eliminate steady-state oscillations and ensure fast convergence. For the deterministic gradients, the authors prove convergence to an exact optimal solution, whereas for stochastic gradients, they establish linear convergence with respect to the iteration number and prove that the convergence error decreases sublinearly with the batch size of sampled data points. The convergence results are reasonable to me.

In my opinion, the most interesting property of the proposed adaptive stepsize approach is that it allows the stepsize to be large in the early stages of the algorithm to accelerate convergence, while rapidly decreasing to a small value near the optimal solution to ensure stable convergence performance. This property appears to provide a "stop signal" for distributed optimization algorithms. If so, it would be useful in real-world distributed optimization applications, as most existing algorithms lack a clear criterion for determining when to stop.

Nevertheless, I have some concerns about this paper, which are summarized below:

(i) Lack of discussion on nonconvex settings (see Weakness 1 for details);

(ii) Limited variety of experimental tasks (see Weakness 2 for details);

**Strengths:**

The paper is well written and easy to follow. The convergence proofs are complete and rigorous. In addition, the proposed adaptive stepsize approch provides an easily identifiable stop signal for distributed algorithms (for example, when the stepsize falls below a given threshold), which is of practical significance.

**Weaknesses:**

1. Lack of discussion on nonconvex settings: Although the experimental results demonstrate the effectiveness of the proposed approach in nonconvex settings, it would be helpful if the authors could elaborate on how the current theoretical results might be extended to nonconvex objective functions.

2. Limited variety of experimental tasks: Aside from logistic regression (in Appendix C), the experiments mainly focus on image classification tasks. To more comprehensively evaluate the effectiveness of the proposed approach, it would be beneficial to test it on other types of machine learning tasks, e.g., distributed training of recurrent neural networks (RNNs) for next-character prediction tasks.

If the authors could address Weaknesses 1 and 2, I would consider raising the score.

**Questions:**

See the weaknesses above. In addition, I have the following questions:

1. In Algorithm 1, the parameter $\beta$ is set as $\beta \in (0, 1.36)$. However, if $\beta < 1$, the stepsize in Algorithm~1 Line 16 would decay exponentially, which may prevent the algorithm from achieving convergence. I also observed that Eq. (43) in Appendix uses $\beta \in (1, 1.36)$. Please clarify the range of $\beta$ to ensure consistency.

2. In the CIFAR-10 experiment, Algorithm 1 does not appear to achieve stable convergence. Therefore, the authors should increase the number of iterations to more clearly show the convergence trend.

3. The use of the iterates $y_{i,2}^{t}$ requires further explanation. It seems to be an additional variable compared with conventional gradient-tracking approaches.

4. There are some typos, e.g.,  (i) under Eq. (7) in Appendix, $||\frac{1}{m}\sum_{i=1}^m a_i||^2 \leq \frac{1}{m}\sum_{i=1}^m ||a_i||^2$ should hold for any vector $a_i \in \mathbb{R}^n$ rather than $a_i \in \mathbb{R}$; and (ii) in Eq.(75), the closing bracket "]'' should be replaced by "\}''.

---

### Official Review · Reviewer_6Yxs · 2025-11-05

**Soundness:** 2
**Presentation:** 3
**Contribution:** 2
**Rating:** 2
**Confidence:** 5

**Summary:**

This paper presents a new algorithm with an adaptive step size for decentralized optimization problems. The authors provide convergence theorems and experimental results for this algorithm.

This paper should be rejected for the following reasons: (1) It contains false theorems and (2) It uses mathematical statements without proof.

**Strengths:**

The authors research the interesting scientific problem - the adaptive step size for decentralized optimization.

**Weaknesses:**

To prove the rejection, I provide the following arguments.

**Main arguments**

1. Theorem 2 contains a false statement about the convergence of the proposed Algorithm ($ \mathbb{E}||x_i^T - x^*||^2 \leq \mathcal{O}(\gamma^T)$, where $\gamma = \max (1 - \frac{\mu^2}{4L}, \frac{91}{92})$), because if this statement were true, we would get any convergence by rescaling. Moreover, this result contradicts standard lower bounds for $L$-smooth and $\mu$-strongly convex functions with $n=1$ and $\sigma = 0$.

2.  P. 16 line 858. The authors used the fact that $L_i^t \leq L$, where $L_i^t$ is defined in Algorithm 1 and $L$ is defined in Assumption 1. Can you provide proof for this fact?

3. P. 20 line 1034. I can't understand this transition. Can you clarify it?

4. p. 20 last inequality. It is not obvious why we can always choose such parameters for all $i$.

5. Also, the relationship (27)  is not obvious.

**Minor comments**

1. How do we define $L_{i}^{t+1}$ if $x_{i}^{t+1} = x_{i}^t$?

2. P. 17 eq. (14). It seems like this equation contradicts Assumption 1.

**Questions:**

I provided questions in the weakness section.

**Things to improve the paper that did not impact the score:**

Provide theoretical guarantees for each statement in this work.

---

> ### Author Response · Authors · 2025-11-27
> **Response to Reviewer 6Yxs**
>
> ${\color{purple} \text{Response to Weakness 1:}} $
>
> We thank the reviewer for pointing out this typo. In the revised manuscript, we have updated $\gamma$ in Theorem 2 from $\gamma=\max(1-\frac{\mu^2}{4L},\frac{91}{92})$ to $\gamma=\max(1-\frac{\mu}{4L},\frac{91}{92})$. We explain that this was a typographical error in the theorem statement, and the proof of Theorem 2 is correct.
>
> ${\color{purple} \text{Response to Weakness 2:}} $
>
> Thank you for the comment. In the revised manuscript, we have removed the inaccurate statement of $L_i^{t} \le L$ in the proof of Lemma 1. Specifically, the condition
> $L_i^{t} \leq L$ was only used to prove the boundedness of $\eta_i^tL_i^t$.  In the revised manuscript, we have refined our proof and established the relationship $\eta_{i}^{t}L_{i}^{t}<\frac{1}{1+a_{2}}(\beta^2+\frac{1}{2})$ (see Eq. (27) to Eq. (34) on page 21 in the revised manuscript for details). Therefore, our updated proof
> ${\color{black} \text{no longer}}$ rely on the condition $L_i^{t} \le L$.
>
> ${\color{purple} \text{Response to Weakness 3:}} $
>
> Yes, we can. In the revised manuscript, we have detailed the proof of Eq. (22) on page 20 (corresponding to the reviewer mentioned P.20 line 1034 in the initial version).
>
> ${\color{purple} \text{Response to Weaknesses 4 and 5:}} $
>
> Thank you for the comment. In Eq. (32) of the revised manuscript, we specify the parameter choices for $a_2, a_5, a_6, a_7$, and $a_8$ (corresponding to $a_2, a_3, a_4, a_5$, and $a_8$ in the previous version) as $a_2\leq\frac{1-r}{4\beta^2}, \max\{a_5, a_6,a_7, a_8\} \leq\frac{47(1-r)}{1600\beta^2}$. Since $r\in(0,1)$ and $\beta\in(1,1.36)$ are fixed global constants, these upper bounds are independent of $i$. Hence we can choose a set of parameters $a_2, a_5, a_6, a_7$, and $a_8$
> satisfying the above bounds **uniformly for all agent $i$**, which guarantees that the last inequality on P. 20 (Eq. (33) in the revised manuscript) holds for every $i$.
>
> Eq. (27) is directly obtained from Step 15 of Algorithm 1, and it is used to ensure that there exist parameters $a_2, a_5, a_6, a_7$, and $a_8$ satisfying the condition in Eq. (33). In the revised manuscript, we further utilize Eq. (27) to establish the key inequality in Eq. (31), which, together with the parameter bounds in Eq. (29), guarantees the validity of the condition in Eq. (33).
>
> ${\color{purple} \text{Response to Minor comment 1:}} $
>
> Thank you for the comment. When $x_i^{t+1} = x_i^{t}$, the value of $L_i^{t+1}$ can be chosen as any positive constant.  For convenience, we set $L_i^{t+1} = 1$ in this case. Therefore, in Line 15 of Algorithm 1, we define $L_{i}^{t+1}=\frac{\|y_{i,1}^{t+1} - y_{i,2}^{t}\|}{\|x_i^{t+1} - x_i^{t}\|}$ if $x_i^{t+1}\neq x_i^{t}$; otherwise, $L_{i}^{t+1}=1$.
>
> ${\color{purple} \text{Response to Minor comment 2:}} $
>
> Thank you for the comment. In our setting,
> $g_i^{t}(x)$ is computed from a minibatch of size $|\mathcal{B}|$: $g_i^{t}(x)= \frac{1}{|\mathcal{B}|}\sum_{j=1}^{|\mathcal{B}|}\nabla \ell(x,\xi_{ij}^{t})$, which leads to a variance bound of $\sigma^{2}/|\mathcal{B}|$. In the revised manuscript, we have corrected the statement in Assumption 1 so that it is now fully consistent with the variance bound $\sigma^{2}/|\mathcal{B}|$ used in Eq. (14).

---

### Meta-Review · Area_Chair_RZDm · 2026-01-06

**Summary:**

This paper studied decentralized distributed stochastic optimization, and proposed a new adaptive stepsize method to solve these decentralized optimization problems. It provided convergence analysis of the proposed algorithm under the strongly convex setting. It also provided some numerical experiments to demonstrate efficiency of the proposed method.

Recently, many works have studied adaptive decentralized optimization and learning. The idea of using adaptive learning rates in decentralized distributed stochastic optimization is not new.  In the paper, many relative papers such as [1,2] are missing.

Although the paper studied convergence properties of the proposed algorithm under the strongly convex setting, clearly, this result is meaningless in practice, e.g. training deep CNNs in the experiments are nonconvex . The authors should provide convergence analysis of the proposed method under the nonconvex setting.  Many comparisons such as [1,2] in the experiments are missing.

[1] Sun, T., Li, D., and Wang, B. Adaptive random walk gradient descent for decentralized optimization. In International Conference on Machine Learning, pp. 20790– 20809. PMLR, 2022.

[2] Chen, X., Karimi, B., Zhao, W., and Li, P. On the convergence of decentralized adaptive gradient methods. In Asian Conference on Machine Learning, pp. 217–232. PMLR, 2023.

**Reviewer Concerns:**

Although the authors have provided rebuttals to address some reviewers' concerns, there exist some concerns still are not addressed. For example, Reviewer WeYr still  believe that the Theorem 1 and 2 are still problematic.

**Reviewer Scores:**

Although the authors have provided rebuttals to address some reviewers' concerns, there exist some concerns still are not addressed. Thus, the likelihood of drastically changing the score is very low.

---

### Decision · Program_Chairs · 2026-01-26

Reject